# Greater Greenland Ice Sheet contribution to global sea level rise in CMIP6

Stefan Hofer [1,2 ✉], Charlotte Lang[2], Charles Amory [2], Christoph Kittel[2], Alison Delhasse[2], Andrew Tedstone[3] & Xavier Fettweis[2]

Future climate projections show a marked increase in Greenland Ice Sheet (GrIS) runoff during the 21st century, a direct consequence of the Polar Amplification signal. Regional climate models (RCMs) are a widely used tool to downscale ensembles of projections from global climate models (GCMs) to assess the impact of global warming on GrIS melt and sea level rise contribution. Initial results of the CMIP6 GCM model intercomparison project have revealed a greater 21st century temperature rise than in CMIP5 models. However, so far very little is known about the subsequent impacts on the future GrIS surface melt and therefore sea level rise contribution. Here, we show that the total GrIS sea level rise contribution from surface mass loss in our high-resolution (15 km) regional climate projections is 17.8 ± 7.8 cm in SSP585, 7.9 cm more than in our RCP8.5 simulations using CMIP5 input. We identify a +1.3 °C greater Arctic Amplification and associated cloud and sea ice feedbacks in the CMIP6 SSP585 scenario as the main drivers. Additionally, an assessment of the GrIS sea level contribution across all emission scenarios highlights, that the GrIS mass loss in CMIP6 is equivalent to a CMIP5 scenario with twice the global radiative forcing.

[1] Department of Geosciences, University of Oslo, Oslo, Norway. [2] SPHERES Research Units, Geography Department, University of Liège, Liège, Belgium. [3] Department of Geosciences, University of Fribourg, Fribourg, Switzerland. ✉email: stefan.hofer@geo.uio.no

The Greenland Ice Sheet (GrIS) has become the largest single source of barystatic sea-level rise[1–5]. Sixty percent of this recent increase in GrIS sea-level contribution is due to enhanced surface runoff[4,6] and GrIS surface processes will also play an important role in a warming climate[2]. Its future mass loss rate strongly depends on the future global temperature rise and therefore anthropogenic greenhouse-gas emission rates[2,7–9], but also on the strength of regional factors such as the melt-albedo feedback[10,11], glacier algae growth[12–15], cloud phase feedbacks[9,16–19], and atmospheric circulation changes[8,20–24]. Global climate models (GCMs) of the Climate Model Inter-comparison Project 5th Phase (CMIP5) show a clear signal of above average temperature rise in the Arctic in various different emission scenarios[25–27]. Regional climate projections of the GrIS melt and near-surface climate unanimously show greater ablation rates with rising Arctic temperature levels[2]. However, the absolute magnitude is still subject to uncertainties, mainly due to imperfect cloud microphysics and missing recent Greenland circulation anomalies[2,8,9,20,28,29].

The latest CMIP 6th Phase (CMIP6) begins to address some of these shortcomings by incorporating more complex model physics, a higher spatio-temporal resolution, and a more realistic coupling between the different Earth system components and better constrained emissions of aerosols and other near-term climate forcers[30–34]. By doing so, the latest GCM model suite is now more sensitive to atmospheric greenhouse gases and therefore shows a stronger temperature increase during the twenty-first century, partly due to stronger cloud feedbacks[30,33–35]. Although CMIP6 models are based on more sophisticated physics and are run at a higher resolution than its predecessor CMIP5, this study does not aim to suggest whether a CMIP6-based world with a greater sensitivity to greenhouse-gas emissions comprises a more likely future scenario than CMIP5-based estimates. However, given the direct connection between global temperature rise, Arctic amplification, and GrIS sea-level contribution, it is of first-order importance to quantify the impacts of this greater Arctic temperature rise in CMIP6 models upon future GrIS melt to assess potential global impacts[36].

Here we show that between the high-emission scenario from CMIP5 (RCP8.5) and CMIP6 (SSP58.5)[31,32,37], which share a similar extreme surface warming of 8.5 W/m² in 2100, GrIS surface melting almost doubles during the twenty-first century. Using a regional climate model (RCM; Modéle Atmosphérique Régional, MAR)[2,38–41]—which explicitly models important polar processes such as the surface mass balance (SMB), snow properties, and radiative transfer—to downscale six CMIP5 RCP8.5 and five CMIP6 SSP585 GCM projections to a higher spatial resolution, we find a cumulative increase of twenty-first century GrIS melt by +28,500 Gt (+7.9 cm sea-level equivalent (SLE) until 2100) and on average a 22 days longer melt season in the CMIP6 simulations. We identify an increase of global temperature of +0.6 °C and even larger increase of +1.3 °C in the Arctic at the end of the twenty-first century as the main driver behind the additional Greenland surface melt in our CMIP6 future projections. Using the statistical connection between Greenland temperature anomalies from GCMs and the annual SMB from our high-resolution simulations, we find a more pronounced GrIS mass loss across all CMIP6 scenarios when compared to CMIP5. Our results highlight that the GrIS could potentially lose ice faster in a warming climate than the previous CMIP5-based estimates suggested.

## Results

**More warming in CMIP6.** The warming rate in CMIP6 SSP58.5 is markedly greater than the warming produced by the CMIP5 RCP8.5 ensemble mean (Fig. 1a). Globally, the 2071–2100 mean temperature increase—compared to the 1961–1990 baseline during which the GrIS had a stable SMB[4]—is +0.6 °C greater in CMIP6 than in CMIP5, despite a similar longwave forcing of 8.5 W/m² in 2100[31,32,37,42]. The difference in temperature increase is especially pronounced in the Arctic, where the deviation between the two GCM ensemble means of the near-surface temperature anomaly for 2071–2100 reaches up to +2.2 °C (Fig. 1a).

The temporal temperature evolution of the CMIP5 and CMIP6 high-emission scenarios reveals that globally CMIP5 and CMIP6 projections start to diverge around 2050, but already in 2030 for the Arctic (>67 °N, Fig. 1b). At the end of the twenty-first century the global mean temperature warming in CMIP6 is +0.6 °C greater, but yields a more than two times greater difference of +1.3 °C for the Arctic, with a notably more sensitive Arctic amplification signal in the CMIP6 GCM ensemble compared to CMIP5.

We chose six CMIP5 GCMs (HadGEM2-ES, MIROC5, NorESM1-M, ACCESS1.3, CSIRO-Mk3-6-0, and IPSL-CM5A-MR—all r1i1p1 ensemble members) and five CMIP6 GCMs (CESM2, CNRM-CM6-1, CNRM-ESM2-1, MRI-ESM2-0, and UKESM1-0-LL - r1i1p1f1 ensemble members, but f2 for CNRM and UKESM) for our dynamical downscaling with MAR. We evaluated all 11 models over the current climate in Supplementary Figs. 1–5 and their equilibrium climate sensitivities are given in Supplementary Table 1 and 2. The mean temporal evolution of the 6 out of 28 CMIP5 GCMs accurately reproduces the overall ensemble mean of all available simulations (Fig. 1c). During the period 1961–1990, the six-model mean was on average −0.15 °C colder than the ensemble mean, and only 0.01 °C warmer during 2071–2100 (Fig. 1c). In addition, the 5 CMIP6 simulations are also representative of the 28 available CMIP6 GCM mean. We selected these five models from the CMIP6 ensemble based on the availability of 6-hourly outputs required for the dynamical downscaling in MAR. During the reference period 1961–1990 our subset was −0.48 °C colder than the average and +0.41 °C warmer between 2071 and 2100, with an overall bias of only −0.13 °C for the whole time series between 1961 and 2100. Therefore, we consider our chosen subset of GCM projections for high-resolution downscaling of the future GrIS SMB to be representative of the overall ensemble means of the CMIP5 and CMIP6 extreme high-emission scenarios.

The absolute temperature increase in the CMIP5 RCP8.5 and CMIP6 SSP585 ensembles shows a strong latitudinal and seasonal variability (Fig. 2a, b), similar to the temperature differences between the two (Fig. 2c). Figure 2a, b both show a clear signal of seasonal variability in temperature feedbacks in RCP8.5 and SSP585 at the end of the twenty-first century (2071–2100). The warming is strongest in both ensembles in winter (+16.9 °C ± 2.9 °C SSP585 vs. +14.9 °C ± 2.9 °C RCP8.5, Fig. 2a, b). In addition, the fall and spring also show a notably enhanced warming signal (Fig. 2a, b). During fall (September, October, November - SON), the mean Arctic warming (>67°N, 2071–2100) is +12.8 °C ± 2.7 °C, whereas it is +8.9 °C ± 1.0 °C in spring (March, April, May - MAM) in RCP8.5, compared to +13.5 °C ± 2.9 °C (SON) and +9.7 °C ± 1.4 °C (MAM) in SSP585. Furthermore, we also see the greatest additional Arctic warming in SSP585 in winter, +2.0 °C more than in RCP8.5, whereas there is a slightly more homogeneous temperature surplus in SSP585 of +0.9 °C on average in spring, summer, and fall (Fig. 2c). Due to the seasonally varying feedback strengths of the albedo-, sea ice-, cloud-, water vapor-, and lapse-rate feedbacks, we conclude that the additional warming in CMIP6 SSP585 can at least be partly explained by a stronger Arctic amplification signal, not only by a uniformly greater warming over all seasons.

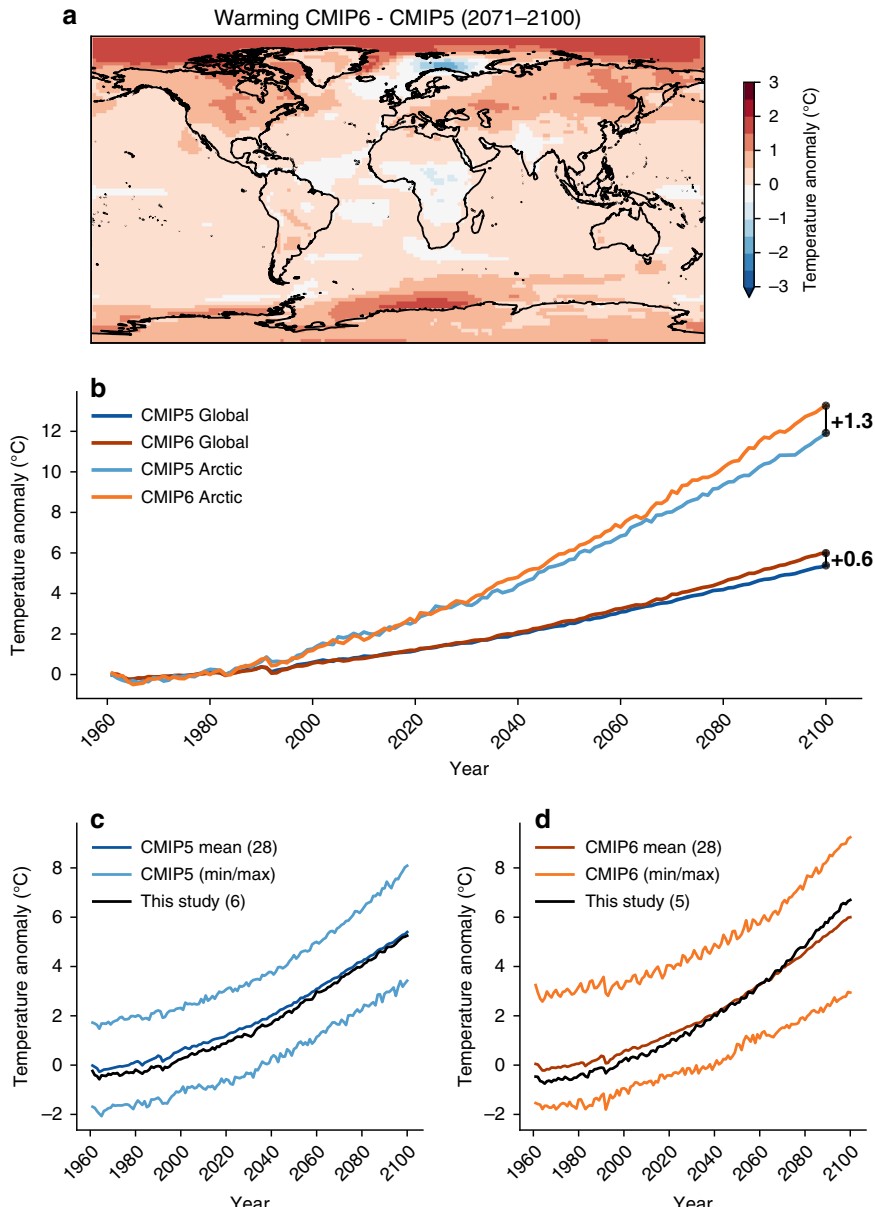

**Fig. 1 Temperature comparison between CMIP5 and CMIP6 models. a** Difference in warming (CMIP6 mean minus CMIP5 mean) at the end of the twenty-first century (2071–2100 30-year average) when compared to the baseline of 1961–1990. Red colors suggest a greater warming in CMIP6. The difference is computed from all available CMIP5 (28) and CMIP6 models (28) from the RCP8.5 and the SSP58.5 scenarios, respectively. **b** Time series of the CMIP5 and CMIP6 global and Arctic near-surface temperature anomaly between 1960 and 2100 (mean of all GCM models), compared to the 1961–1990 baseline. **c** Comparison between the 28 model CMIP5 global mean temperature anomaly (dark blue, reference period 1961–1990) and the mean of the 6 GCMs chosen for our dynamical downscaling (black), and the minimum and maximum of all CMIP5 GCMs (pale blue). **d** Same as **c** but for 28 CMIP6 GCMs and the mean of the 5 GCMs chosen for our downscaling in this study.

**Higher Greenland mass loss in SSP585 than RCP8.5.** There is a stark contrast in future GrIS surface melt and meltwater runoff between the two model suites (Fig. 3). At the end of the twenty-first century, MAR SSP585 projects a doubling in the reduction of GrIS SMB compared to the CMIP5 (RCP8.5) forced MAR simulation. Between 2091 and 2100, the average annual SMB is −1332 Gt/yr in the SSP585 10-year average, compared to −456 Gt/yr under RCP8.5 (Fig. 3a). This difference in SMB is equivalent to a 2.92 times increase in future GrIS surface mass loss rate at the end of the twenty-first century under SSP585.

The surplus in SSP585 GrIS surface mass loss only becomes notable from 2020 onwards. During the period of stable SMB 1961–1990, the SSP585 simulations on average have a +26 Gt/yr greater annual SMB, a difference of around 5% when compared to the annual SMB. Conversely, at the end of the century (2071–2100 average) the SMB is −602 Gt/yr lower in our SSP585 simulations compared to the RCP8.5 simulations, with the differences reaching up to 292% during 2090–2100. Therefore, the SSP585 simulations acquire their differences during the course of the twenty-first century. This is also highlighted by our comparison of MAR forced by reanalysis to each individual MAR simulation forced by GCMs over the current climate in the Methods section (Supplementary Figs. 1–5). Supplementary Figs. 1–5 show that over the current climate the SMB and other variables in MAR forced by reanalysis compared to GCM forcing do not differ significantly. Over the current climate (1981–2010),

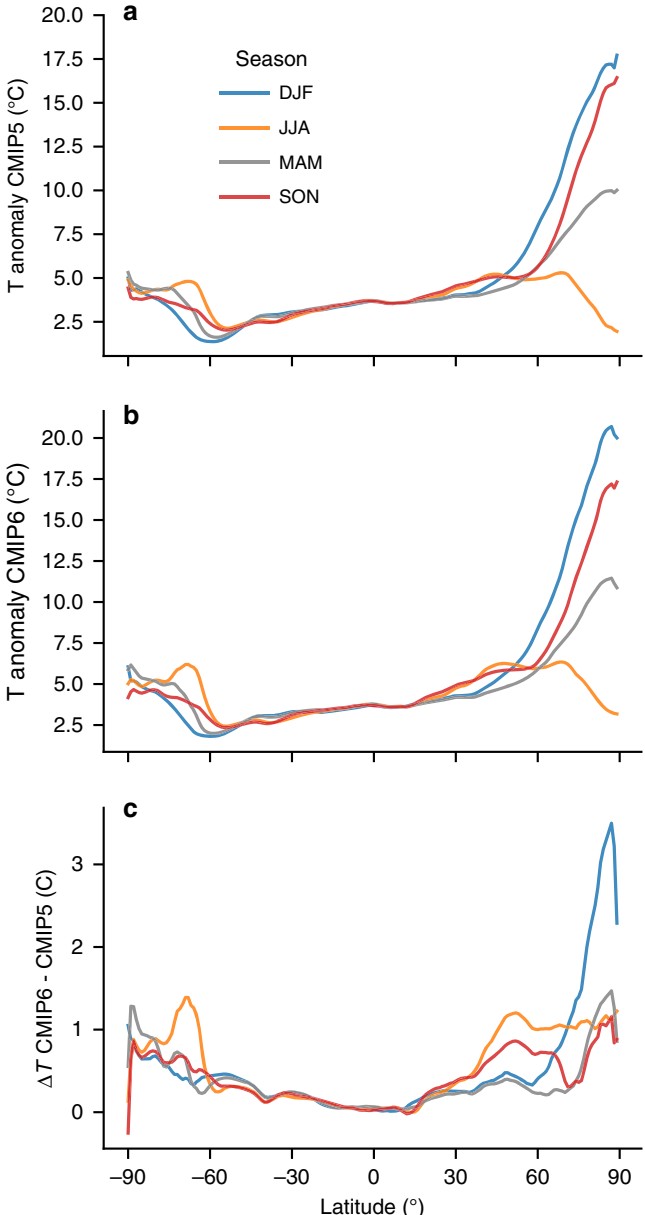

**Fig. 2 Seasonality of warming in the CMIP5 and CMIP6 extreme high-emission scenarios. a** CMIP5 2 m temperature anomaly (RCP8.5, $n = 28$) from south to north for the period 2071–2100, compared to the baseline climate of 1961–1990. Different lines represent the warming in four different seasons. **b** Same as **a** but for the CMIP6 SSP585 scenario ($n = 28$). **c** Difference in warming in 2071–2100 between CMIP6 (SSP585) and CMIP5 (RCP8.5) (positive = more warming in CMIP6).

the SMB in MAR forced by SSP585 models does not show an inherent bias when compared to the MAR RCP8.5 simulations (mean bias: 29.8 mm w.e. (SSP585) compared to 26.3 mm w.e. (RCP8.5)). Therefore, the most likely explanation for greater GrIS surface mass loss is the higher equilibrium climate sensitivity (ECS) in CMIP6 models and a more pronounced Arctic amplification (Figs. 1 and 2)[30,33–35].

In cumulative terms (Fig. 3b), the mean of the CMIP6 SSP585 MAR simulations projects twenty-first century SMB anomalies of −64,676 Gt ± 28,100 Gt (+17.8 cm ± 7.8 cm SLE[43]), whereas the CMIP5 RCP8.5 simulations show a less intense surface mass loss of −36,210 Gt ± 22,000 Gt (+9.9 cm ± 6.0 cm SLE). As both SMB means only start to diverge around 2020, the cumulative SMB

anomalies differ by a factor of 1.8 (greater cumulative SMB reduction in CMIP6 simulations), lower than the factor of 2.92 for annual SMB at the end of the twenty-first century (Fig. 3a, b).

The statistical comparison between the global temperature anomalies of our subset of CMIP5 and CMIP6 GCMs, and their downscaled Greenland SMB shows that for a given global temperature anomaly, the SMB is notably lower in our SSP585-forced MAR simulations than when forced by RCP8.5 from CMIP5 (Fig. 4a). Although the second-order polynomial fit is similar until the global temperature anomaly in the GCMs reaches +2 °C, they start to diverge under greater warming. Conversely, when considering the Greenland meltwater production and the local (60°N–85°N, 20°W–70°W) 600 hPa JJA (June, July, August) temperature anomaly, the statistical connection between temperature and melt anomalies is almost identical between CMIP5 and CMIP6 simulations (Fig. 4b). This similarity points towards that the local feedbacks between temperature and melt during melt season—e.g., a stronger melt-albedo feedback over the ice sheet are not the main cause of greater melt in the CMIP6 MAR simulations (Fig. 4b). Instead, the main driver appears to be a stronger Arctic temperature amplification in CMIP6 when averaged over our (downscaled) ensemble. However, one also needs to consider that temperature anomalies at 600 hPa do not only explain temperature variability. The 600 hPa temperature is also closely connected to the atmospheric circulation state (i.e., higher mid-tropospheric temperatures in anticyclonic circulation) and therefore are also correlated to the downwelling radiative fluxes (e.g., more solar radiation in anticyclonic conditions) and potential radiative feedbacks.

Further, although summer melt dynamics explain most of the variability in annual SMB over Greenland, they do not account for the entire range of physical processes involved. Therefore, Fig 4c shows the statistical connection between the local temperature anomaly in our five CMIP6 SSP585 and six CMIP5 RCP8.5 GCMs (60°N–85°N, 20°W–70°W), and the annual SMB over Greenland from our downscaled RCM projections. The second-order fit shows that (i) for a given temperature anomaly the annual GrIS SMB is lower in CMIP6, (ii) that in CMIP5 the threshold where surface mass loss outweighs snow accumulation (i.e., SMB lower than 0) will be reached at +4.4 °C, and (iii) that the same threshold will already be reached at +3.7 °C in CMIP6. In addition, the variability between individual model realizations in Fig. 4c also clearly shows that studies relying on only one downscaled GCM for statistical upscaling of GrIS mass loss across all Representative Concentrations Pathway (RCP) and SSP scenarios are potentially prone to significant biases compared to a larger downscaled ensemble.

**Greenland surface mass loss across multiple emission scenarios.** Using the statistical relationship between the GrIS SMB and the local temperature anomalies (Fig. 4c), we can reconstruct the twenty-first century GrIS SMB evolution across multiple CMIP5 and CMIP6 emission scenarios and ensemble members.

Overall, it is notable from all scenarios that Greenland will lose more mass and at a faster rate in CMIP6 than in CMIP5 (Fig. 5), in line with our RCM simulations. In the extreme emission scenarios RCP8.5 and SSP585 (Fig. 5a), the mean SMB-related GrIS mass loss is 2.2 times greater in CMIP6 than in CMIP5 (−942 Gt/yr vs. −430 Gt/yr). In addition, the surface ablation already outweighs the snow accumulation (i.e., SMB < 0) in 2046 in SSP585, compared to 2058 in RCP8.5. The SMB in RCP8.5 and SSP585 already starts to deviate around 2020, indicative that the onset of the greater atmospheric warming around Greenland in SSP585 lies in the early twenty-first century. Similar results are evident for lower emissions scenarios (Fig. 5b, c). In SSP245, the

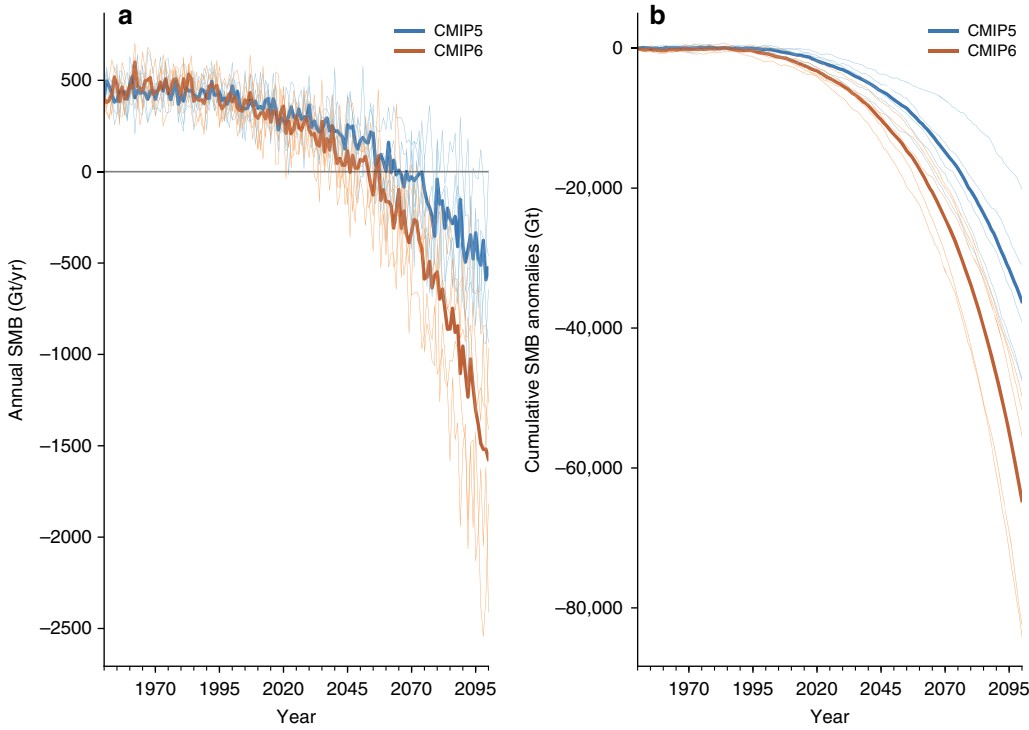

**Fig. 3 Greenland surface mass balance comparison between downscaled CMIP5 and CMIP6 MAR simulations. a** Annual Greenland Ice Sheet wide integrated annual SMB from six CMIP5 forced MAR simulations (blue) and five CMIP6 MAR simulations (orange) in Gigatonnes per year (Gt/yr). The dark blue line represents the mean of all CMIP5 MAR simulations (6, RCP8.5) and dark orange is the mean of all CMIP6 MAR simulations (5, SSP585). The individual runs are shown in lighter colors. **b** Same as **a** but for the cumulative GrIS SMB anomalies (Gt), based on the 1961–1990 average of the simulations.

mean SMB drops below 0 in 2066, whereas the mean of the RCP45 scenario stays (just) above 0 Gt/yr throughout the twenty-first century (2100: −164 Gt/yr vs. +33 Gt/yr). Although in the lowest emission scenarios (RCP2.6, SSP126) both ensemble means stay above the SMB threshold, SSP126 still shows a 120 Gt/yr lower SMB at the end of the twenty-first century than its CMIP5 counterpart (SSP126: +105 Gt/yr, RCP2.6: +225 Gt/yr). Overall, the reconstructed mass loss in CMIP6 is comparable to an emission scenario with twice the surface warming in CMIP5 (SSP126–>RCP4.5, SSP245–>RCP8.5, and SSP585–>NA).

From the twenty-first century SMB reconstruction across all CMIP5 and CMIP6 scenarios, we can also reconstruct the overall GrIS SMB contribution to global sea-level rise from all emission scenarios (Table 1 Column 1, SLE [cm][43]). Overall, the GrIS SLE until 2100 through surface processes varies between a best case estimate of 5.2 cm ± 1.8 cm in RCP2.6 and a worst case esimate of 16.0 cm ± 7.4 cm in SSP585. Across all SLE estimates, it is notable again, that the CMIP6 GrIS sea-level rise contribution is roughly equivalent to a scenario with twice the global surface radiative forcing in CMIP5. For example, the mean SLE of the SSP126 scenario is 7.8 cm ± 4.1 cm contribution to global sea-level rise, while the RCP4.5 scenario with 1.9 W/m² additional global warming yields a comparable 7.4 cm ± 2.1 cm of GrIS SLE. Furthermore, Table 1 second column shows that our subset of physically downscaled RCM simulations project a comparable twenty-first century SLE evolution as when considering all CMIP ensemble members. In RCP8.5 our six RCM simulations lead to an SLE of 9.9 ± 6.0 cm compared to 11.0 cm ± 3.1 cm when considering all CMIP5 members. In SSP585, our five RCM simulations also show comparable results with an SLE of 17.8 cm ± 7.8 cm, whereas the statistical reconstruction across all CMIP6 SSP585 models would lead to 16.0 cm ± 6.4 cm of GrIS sea-level contribution. This similarity is another indication that our chosen

subset of GCM models for physical downscaling with MAR is a representative subset of the overall ensemble.

**Physical drivers of greater Greenland mass loss.** The RCM simulations show that the main driver behind the more pronounced decrease in SMB in CMIP6 SSP585 is a greater surface melt and subsequent meltwater runoff (Fig. 6a, b) . Overall, the cumulative melt anomalies are +59,292 Gt in RCP8.5 MAR simulations and +90,757 Gt in SSP585. In both downscaled CMIP averages, the increase in surface melt is only partly offset by an increase in snowfall (+7236 Gt RCP8.5, +4545 Gt SSP585) and the increase in snowfall during the twenty-first century is 38% lower in SSP585, partly explaining the difference in SMB for a given surface warming between CMIP5 and CMIP6 (Fig. 4c). In addition, due to the greater increase in warming in SSP585, converting solid to liquid precipitation, there is a larger increase in rainfall over the GrIS, of +47% compared to RCP8.5 (5974 Gt vs. 8782 Gt), which can have notable implications upon surface albedo[11].

There are also notable differences in the duration and intensity of the ablation seasons (Fig. 6c). Although such differences between the RCP8.5 (CMIP5) and SSP585 (CMIP6) MAR simulations are negligible during 1981–2010, differences develop by the end of the twenty-first century (2071–2100). First, the peak daily ablation intensity in the RCP8.5 simulations is −15 Gt/day, whereas in SSP585 it is −23 Gt/day, an increase of 50%. Second, the ablation season in the SSP585 simulations, defined as daily SMB < 0 Gt/day, starts 7 days earlier and ends 15 days later. The GrIS therefore experiences surface mass loss on 22 additional days in the CMIP6 simulations, which is asymmetric, extending longer at the end of the melt season in SSP585, when the melt is mostly driven by incoming longwave radiation due to already lower solar radiation. However, most of the extra mass loss in

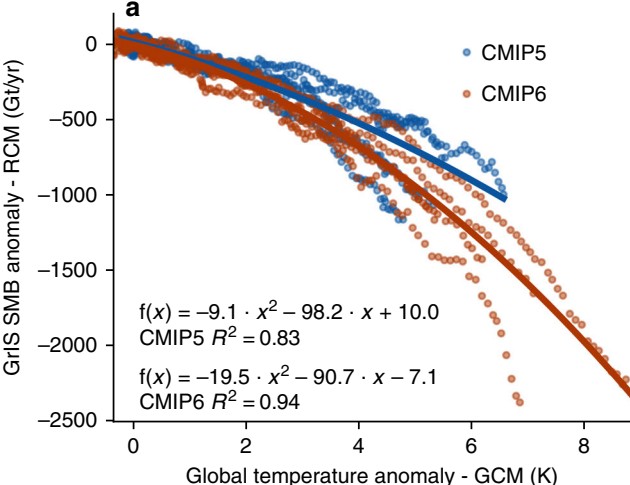

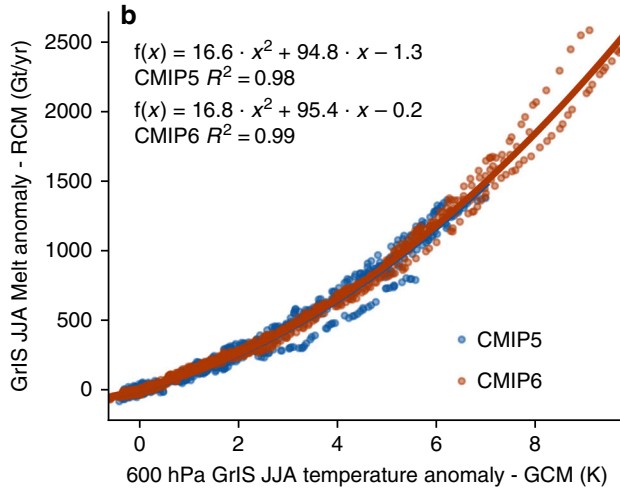

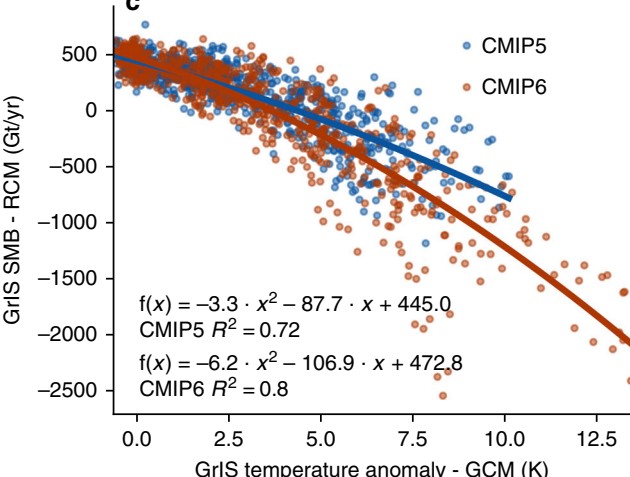

**Fig. 4 Statistical comparison between GCM temperature anomalies and GrIS melt and SMB anomalies from regional climate simulations.**
**a** Scatterplot of 10-year running means of the annual global temperature anomalies from the six CMIP5 (RCP8.5) and five CMIP6 (SSP585) GCM models (x-axis) and the dynamically downscaled MAR GrIS SMB anomalies (y-axis) in gigatonnes (Gt). The orange and blue lines are second-order polynomials fitted to the data ("least squares"), where the function is given in the figures alongside the $R^2$ correlation scores. The anomalies are computed for the 1961–1990 mean state. **b** Same as **a** but for the comparison between 600 hPa JJA temperature anomaly over the Greenland domain (60°N–85°N, 20°W–70°W) vs. the GrIS melt anomaly during summer (JJA) in gigatonnes (Gt). **c** Same as **b** but for the comparison between the Greenland (60°N–85°N, 20°W–70°N) 2 m annual temperature anomaly (GCMs) and the annual GrIS surface mass balance (RCM).

by +7.9 cm in CMIP6 MAR simulations, we also analyzed the difference in the corresponding GrIS radiative surface energy budget (SEB) fluxes (Fig. 7, 2071–2100). Overall, the greatest differences in SEB fluxes occur during the summer melt season (Fig. 7, third row). During summer, the shortwave downward fluxes (SWDs) are lower in our five SSP585 MAR simulations than in our six RCP8.5 simulations, with a spatial focus on the northern and western periphery of the GrIS (Fig. 7), concentrated in the ablation areas ($10 \times 10$ grid cell mean: $\approx -3$ W/m$^2$ in southwest, $\approx -5$ W/m$^2$ in northwest ablation zones). Conversely, over the interior and the southeastern GrIS, SWD is slightly enhanced ($10 \times 10$ grid cell mean: $\approx +4$ W/m$^2$ interior, $+8$ W/m$^2$ in southeast). Despite this decrease in SWD over large areas of the ablation zone, the main driver of the greater summer melt in our CMIP6 simulations still is an increase in absorbed shortwave fluxes (SWnet) over all of the ablating areas during summer (Fig. 7, SWnet). A direct consequence of the melt-albedo feedback[10]. Further, the incoming longwave fluxes (LWDs) are enhanced in all seasons due to the greenhouse-gas-driven temperature increase and Arctic Amplification[44,45]. However, the LWD increase is strongest in JJA and in areas where the SWD decrease is greatest, indicative of a cloud optical depth and atmospheric emissivity effect as discussed in ref. [9].

One of the potential hypotheses for a greater cloud optical depth during summer melt season towards the end of the twenty-first century is a greater loss of sea ice in the five CMIP6 SSP585 models we chose for downscaling (Fig. 8). The mean summer location of the southern edge of the continuous sea ice at the end of the twenty-first century (2071–2100) in our five CMIP6 models in the Baffin Bay sector migrates ~1000 km further north along Baffin Island and moves roughly 400 km north in the northwest of Greenland compared to our 6 CMIP5 models (Fig. 8). Conversely, in the Greenland Sea sector east of the GrIS the southern sea ice edge lies roughly 600 km north where it is connected to Greenland, but less so over the open ocean. In general, in our SSP585 models only the northernmost edge of Greenland, where most of the multi-year sea ice resides, is still covered in sea ice between 2070 and 2100. Apart from the north, Greenland will be surrounded by open water and a quasi-unlimited moisture and heat source in SSP585 at the end of the twenty-first century during summer. Although higher moisture availability likely explains part of the increase in cloud optical depth seen from the SEB analysis in Fig. 7, we cannot assess from our data how much of the increase in cloud optical thickness is due to a phase change from mostly ice-containing clouds to mixed-phase or liquid clouds[9,16,17]. In addition, recent studies using lagrangian moisture tracking also point towards that a notable proportion of humidity is advected to the GrIS on the

SSP585 occurs due the greater ablation intensity. Conversely, during the accumulation period, we find no noticeable differences between the RCP8.5 and SSP585 MAR simulations, leading to the conclusion that melt season dynamics are the main driver of the divergent SMB projections between CMIP5 and CMIP6 simulations.

To assess where the surplus in energy is coming from to increase the twenty-first GrIS melt and sea-level contribution

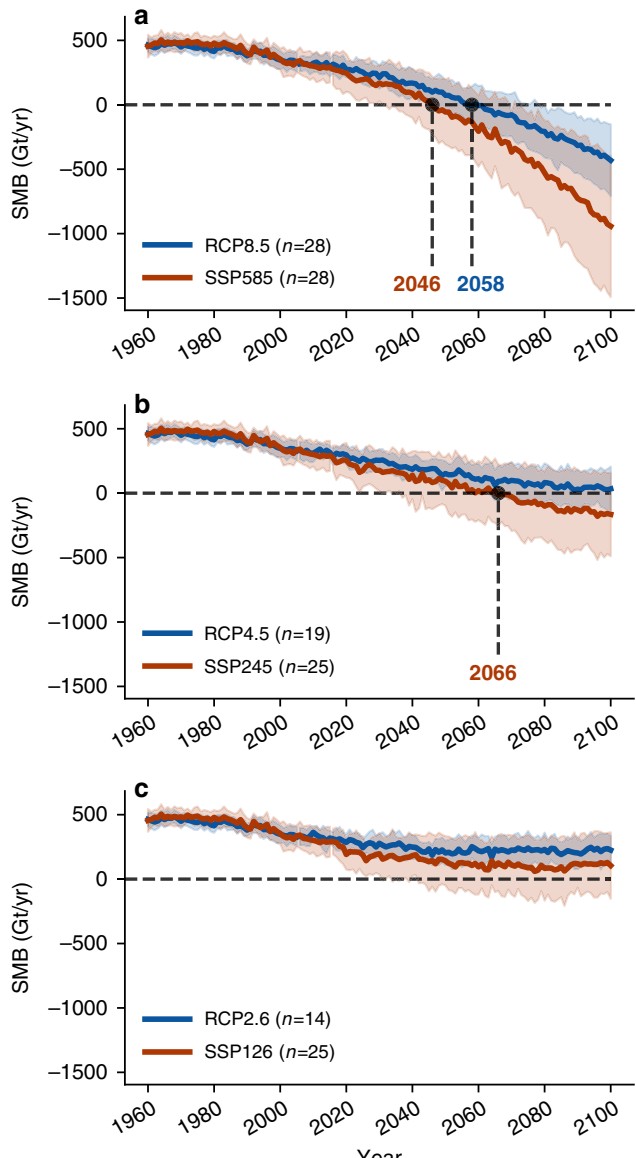

**Table 1 Overview of the twenty-first century Greenland sea-level rise contribution from surface mass loss across all CMIP5 and CMIP6 scenarios.**

| Scenario | Reconstructed SLE [cm] | MAR SLE [cm] | Threshold SMB < 0 [yr] |
|---|---|---|---|
| RCP2.6 | 5.2 [4.4–7.0] | NA | [-] |
| SSP126 | 7.8 [3.7–11.9] | NA | [2042-] |
| RCP4.5 | 7.4 [5.3–9.5] | NA | [2063-] |
| SSP245 | 10.2 [5.4–15.0] | NA | 2066 [2037-] |
| RCP8.5 | 11.0 [7.9–14.1] | 9.9 [3.9–15.9] | 2058 [2048–2079] |
| SSP585 | 16.0 [9.6–22.4] | 17.8 [10.0–25.6] | 2046 [2032–2070] |

The "Reconstructed SLE" column denotes the 1961–2100 surface mass balance contribution to global sea-level rise from Greenland in cm—reconstructed from the CMIP5 and CMIP6 GCMs—with the ±1 SD range given in brackets. The "MAR SLE" column shows the same variable but from the phyically downscaled six CMIP5 and five CMIP6 MAR simulations. The last column shows the year in which the mean of the CMIP5 and CMIP6 ensemble drop below 0 Gt/yr, with the values in brackets denoting the year in which the mean ±1 SD range drops below 0 Gt/yr.

**Fig. 5 Reconstruction of the Greenland Ice Sheet surface mass balance across all CMIP5 and CMIP6 emission scenarios. a** Annual Greenland Ice Sheet surface mass balance (Gt/yr) for the CMIP5 and CMIP6 extreme high-emission scenarios (RCP8.5 and SSP585), reconstructed with the individual equations found in Fig. 4c). The orange and blue numbers denote the years in which the mean SMB of the CMIP6 and CMIP5 ensemble drops below 0 Gt/yr, the colored lines represent the ensemble mean, the shaded areas cover ±1 SDs around the mean. The size of the ensemble is given in the figure legend (e.g., *n* = 28). **b** Same as **a** but for the RCP4.5 and SSP245 scenarios. **c** Same as **a** and **b** but for the low-emission scenarios RCP2.6 and SSP126.

synoptic-scale and not locally from the surrounding ocean[46], which will have to be evaluated in more detail in future studies.

## Discussion

Our study highlights that sea-level contribution from the GrIS could be larger and faster than previously thought. Our CMIP6-based extreme high-emission scenario (SSP585) RCM simulations suggest a barystatic GrIS sea-level contribution of +17.8 cm, 7.9 cm more than in our CMIP5-based RCP8.5 model simulations. Under the new SSP585 scenario, RCM-based estimates of future

GrIS SMB results in 80% more twenty-first century surface mass loss and barystatic sea-level rise contribution than when using its CMIP5 predecessor RCP8.5. The mass loss rate at the end of the twenty-first century is almost tripled (x2.92) with SSP585 forcing. We identify a stronger Arctic amplification signal in the CMIP6 SSP585 ensemble—together with associated sea ice (Fig. 8) and radiative (cloud) feedbacks (Fig. 7)—as the main drivers behind the increase in GrIS melt and SMB reduction, which emphasizes the need for realistic representations of high-latitude climate physics in state-of-the-art GCMs. However, first analysis suggests, despite more sophisticated physics in CMIP6, that in individual models (CESM2) of the CMIP6 ensemble the climate sensitivity might not be in-line with paleoclimate records (i.e., too high)[48]. A similar analysis has not been done across all CMIP6 models and therefore a thorough assessment of the models' ability to realistically reproduce paleoclimatic warming periods cannot be assessed at this stage.

Our results also provide an estimate of the Greenland contribution to global sea-level rise across other emission scenarios. Using the statistical connection between Greenland temperature rise in GCMs and the surface mass loss from our high-resolution simulations (Fig. 4c), we also find a greater mass loss across all other CMIP6 emission scenarios when compared to CMIP5 scenarios with a similar radiative forcing (Table 1). Overall, the CMIP6 Greenland mass loss in a given scenario is comparable to the mass loss in a CMIP5 scenario with approximately twice the global warming from anthropogenic emissions. Our study indicates that the twenty-first century sea-level contribution from Greenland is on average 2.6 cm higher in the CMIP6 scenarios in a low-emission future (RCP2.6, SSP126), 2.8 cm higher in a medium-emission future (RCP4.5, SSP245), and 5 cm more in the extreme high-emission scenarios (RCP8.5, SSP585), despite a similar level of global radiative forcing. Further, the year in which the GrIS will lose more mass from meltwater runoff than it gains from snow accumulation in winter could be reached 12 years sooner in SSP585 compared to RCP8.5, whereas this potential tipping point would be crossed in SSP245, but not in RCP4.5.

We note that our RCM simulations currently assume a fixed ice sheet geometry (present day), which underestimates the positive melt-elevation feedback but also does not take into account the ice sheet extent- and shape-melt feedbacks. However, a previous coupled MAR-ice sheet model setup using a fixed ice sheet extent has shown a negligible overestimation of the GrIS sea-level rise contribution, by only 1% in 2100 and 6% in 2150 in one

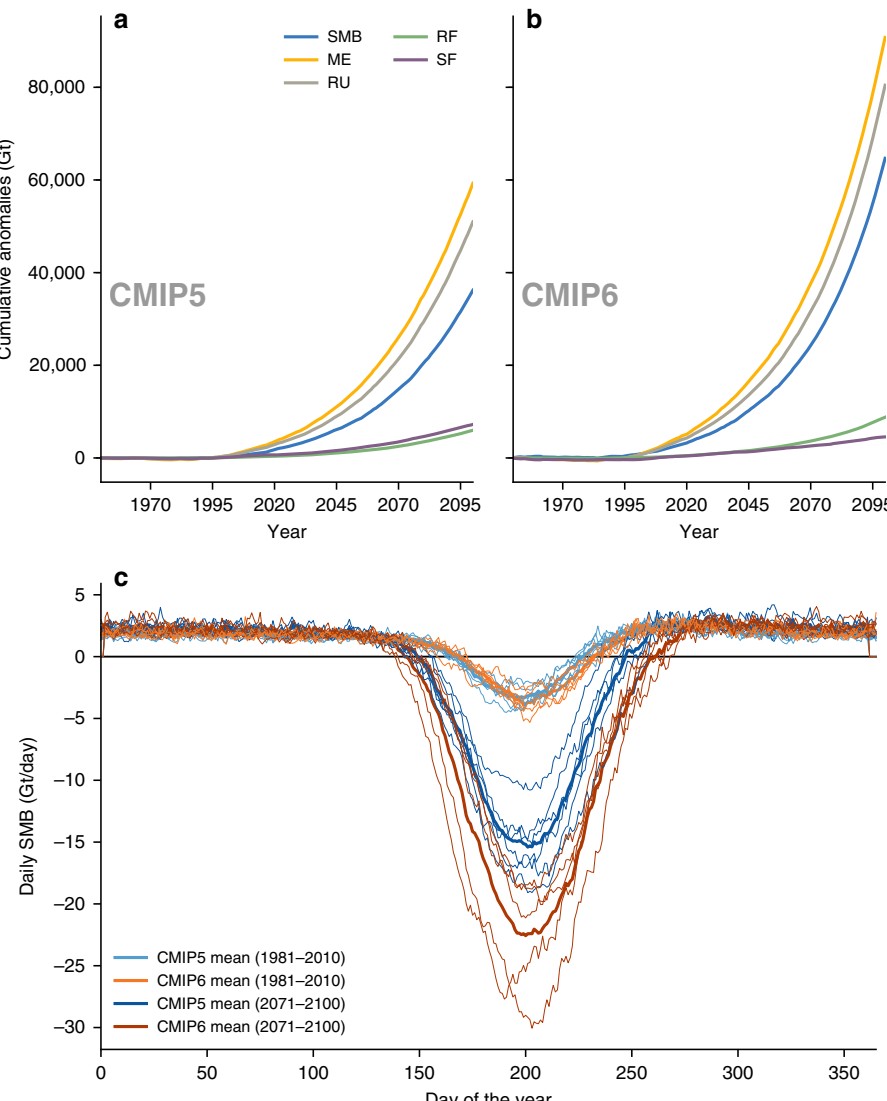

**Fig. 6 Greenland Ice Sheet surface mass balance components and annual surface mass balance cycle in CMIP5 and CMIP6. a** Cumulative SMB component anomalies from the mean of all six CMIP5 (RCP8.5) MAR simulations based on the 1961–1990 mean of the simulations. "ME" corresponds to the cumulative melt-, "RU" to the runoff-, "RF" to the rainfall-, and "SF" to the cumulative snowfall anomalies—all in Gigatonnes (Gt). Note: the negative SMB anomalies are multiplied by -1 for better visible comparison with positive melt anomalies. **b** Same as **a** but based on the mean of all five CMIP6 SSP585 MAR simulations. **c** Annual SMB cycle (Gt/day) from CMIP5 RCP8.5 (blue) and CMIP6 SSP585 (orange) forced MAR simulations for two different 30-year periods (1981–2010 and 2071–2100). The thicker lines represent the CMIP5 and CMIP6 MAR simulation mean, whereas the individual simulations are shown in thinner lines.

RCP8.5 simulation[49]. Conversely, CMIP6-based melt estimates presented in our study could also be too low, given that both CMIP5 and CMIP6 models currently lack the ability to accurately reproduce recent Arctic general circulation anomalies, which have been responsible for recent record mass losses and could be a source of greater twenty-first century GrIS mass loss[8,9,20].

Our study highlights multiple far-reaching implications for science and policy-makers. (i) Policy-makers should consider using multiple lines of evidence when planning future mitigation and adaptation efforts, because of the rapidly improving representation of physical processes in climate models and their impact on sea-level rise estimates. (ii) Future GCM development might prove to be one of the leading sources of uncertainties in future global sea-level rise estimates. In our study, by using the latest suite of CMIP6 SSP585 models, we force our RCM with data from climate models that incorporate the most up-to-date atmospheric parameterizations and that have also significantly increased the

spatio-temporal resolution of the driving GCMs. This change in complexity from CMIP5 RCP8.5 to SSP585 has led to a doubling in the GrIS barystatic sea-level contribution in our study by +7.9 cm, despite the same background forcing from greenhouse gases (+8.5 W/m²). In addition, we also find that the GrIS contribution is greater throughout all CMIP6 emission scenarios when compared to CMIP5, despite the same radiative forcing from anthropogenic emissions.

## Methods
**Climate Model Intercomparison Project 5th Phase**. The CMIP5 was designed as a climate model intercomparison initiative to address some of the pressing questions for the Fifth Assessment Report (AR5) of the IPCC[1,25,37]. For the first time in the CMIP projects, atmosphere-ocean general circulation models (AOGCMs) were coupled to dynamic biogeochemical components, allowing to close the terrestrial carbon cycle and creating the first Earth system models (ESMs)[37,42]. In comparison to CMIP6, where the future scenarios were removed from the core experiments, CMIP5 addressed both the historical simulations and the future scenarios in its

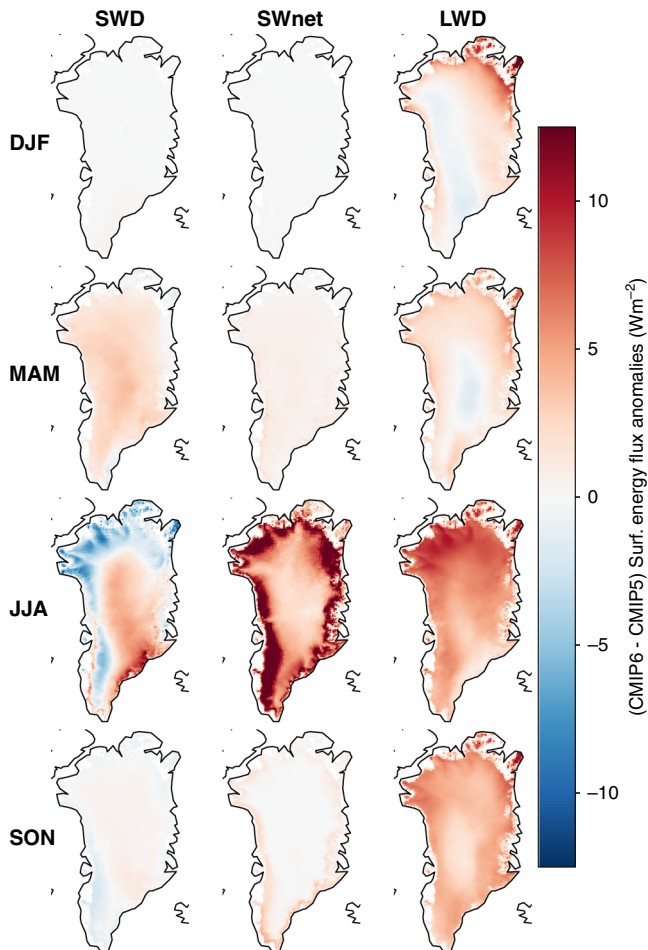

**Fig. 7 Difference in surface energy budget fluxes between CMIP6 and CMIP5 MAR simulations (2071–2100).** Thirty-year average (2071–2100) difference SSP585–RCP8.5) in incoming longwave and shortwave radiation, in addition to the absorbed shortwave radiation difference in W/m². Positive values indicate a greater downwelling or absorbed flux in CMIP6 SSP585 (red colors), while negative values indicate lower fluxes towards the surface in CMIP6 (blue colors). The rows indicate the difference seasons, from winter (DJF) at the top, spring (MAM), summer (JJA) and fall at the bottom (SON). In addition, the columns indicate the different components of the radiative fluxes, namely incoming shortwave (SWD), absorbed shortwave (SWnet), and incoming longwave radiation (LWD).

main objectives[31,32,37]. However, out of the four RCPs only RCP4.5 and RCP8.5 were mandatory for participation in CMIP5[37]. In this study we mainly rely on two core experiments of CMIP5, namely the historical simulations and the high-emission scenario RCP8.5[37,42].

Here we used the historical (1850–2005) simulation of CMIP5 and the RCP8.5 high-emission future scenario (2006–2100)[37,42]. The historical simulation we used was forced from observed atmospheric composition changes (e.g., greenhouse gas and aerosol emissions), solar forcing and also included information about observed land-use changes[37]. Because of these observational constraints on the solutions in CMIP5 AOGCMs and ESMs, the period 1850–2005 allows for a detailed characterization of modeled biases in essential climate parameters in the CMIP5 ensemble members[37].

For our downscaled CMIP5 future simulations, we use the extreme high-emission scenario RCP8.5 (2006–2100)[37,42]. In this scenario, rapid and fossil-fuel based development leads a CO2-equivalent concentration of >1370 p.p.m. in the year 2100[42]. This significant increase in atmospheric CO2-levels, in addition to changes in land use and land cover, leads to a radiative forcing of +8.5 W/m² at the end of the twenty-first century and rapidly increasing atmospheric temperature levels[42].

**Climate Model Intercomparison Project 6th Phase.** The CMIP6 is the latest and ongoing climate model intercomparison initiative[31,32]. For modeling groups to take part in CMIP6 they need to comply with two main scientific purposes: (1) a

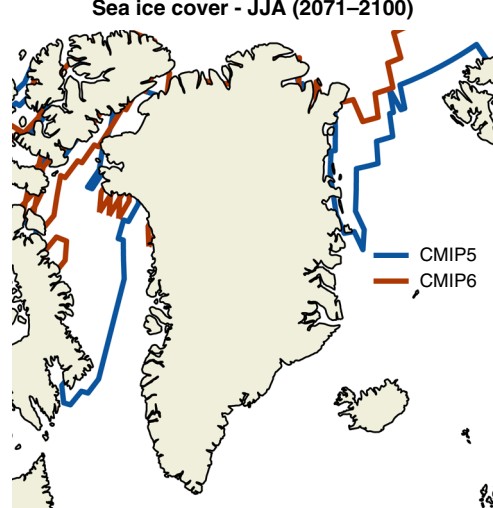

**Fig. 8 Difference in JJA sea ice coverage between 5 CMIP6 and 6 CMIP5 models used in the downscaling with MAR.** Thirty-year average (2071–2100) location of the southern edge of the continuous Arctic sea ice cover around Greenland in JJA—computed as the boundary where the sea ice concentration ("sic" or "siconc" variable) of the GCMs is greater than 15% as used in ref. [47].

"handful of common experiments"[31] and (2) apply "common standards"[31] in terms of model documentation, infrastructure and data publication[31]. In addition, there is a third foundation on which CMIP6 heavily relies, which are the 21 CMIP6-Endorsed MIPs (e.g., GeoMIP and ScenarioMIP). Participation in these is not a mandatory requirement[31]. In this study, we mainly rely on one core experiment, namely the historical simulations, but also on output of one MIP, namely the ScenarioMIP[31,32].

Throughout this study we mainly make use of the historical simulations (1850–2014) of the CMIP6 core experiments and the ScenarioMIP high-emission scenario SSP58.5 (2015–2100)[31,32]. The historical simulation of CMIP6, very similarly to CMIP5, prescribes observed short- and long-lived greenhouse gas emissions[31]. In addition, for AtmosphereMIP simulations, the sea surface temperature and sea ice concentration are prescribed based on observations[31]. These observational constraints, which all the CMIP6 models share, allows for a comparison of systematic and model-specific inherent biases[31].

For the future simulations in this study we use the rapid-development but fossil-fuel based Shared Socioeconomic Pathway58.5 scenario (2015–2100)[32]. In terms of social development, SSP5 represents a scenario in which rapid economic growth leads to stable institutions and large investments in education[32]. Although this is also true for SSP1, the growth in SSP5 relies heavily on fossil-fuel extraction and therefore leads to a greater global average climate forcing of +8.5 W/m² in 2100[32]. The SSP58.5 scenario, alongside the other three "Tier-1" scenarios, has been designed as an updated version of the RCP8.5 from CMIP5[31,32,42], allowing for a comparison of two sets of simulations, despite small differences in emissions and land use[32].

**Difference between CMIP6 and CMIP5 scenarios.** The ScenarioMIP SSP58.5 scenario was defined as an updated version of the RCP8.5 scenario, but the two scenarios slightly differ in the way they achieve a net global average forcing of +8.5 W/m² in 2100[32]. In CMIP6 SSP58.5, the underlying assumptions are based on new emission- and land-use scenarios[50]. For example, CMIP5 RCP8.5 and CMIP6 SSP58.5 differ by roughly 100 p.p.m. of atmospheric CO2 concentration in 2100 [32] (Fig. 3), while the time evolution of total anthropogenic radiative forcing and twenty-first century temperature rise are almost identical[32] (Fig. 3).

This similarity in essential climate variables despite higher CO2 concentration is achieved by compensating influences in the projected land use (cropland, pasture, forests) and near-term climate forcers (e.g., aerosols,[32]). Thus, although globally RCP8.5 and SSP58.5 should lead to similar climate forcing and temperature evolutions, the two scenarios can differ on regional and especially local scales[32].

In addition, the models in CMIP6 use updated model physics and higher spatio-temporal resolution than their CMIP5 predecessors, leading to the question of how much of the differences between RCP8.5 and SSP58.5 can be attributed to updated climate models and how much can be due to local differences in climate forcing? Initial results of the notably greater ECS in CESM2 highlights that more sophisticated cloud microphysics are the main contributor to the greater climate sensitivity[30]. Although this is an interesting result, the fact that such an analysis is yet to be done for most other CMIP6 ensemble members means that the research

on the underlying causes of the greater twenty-first temperature increase in CMIP6 compared to CMIP5 for the same radiative forcing is still emerging. However, using an unchanged climate model (MAGICC version 6.8.01),[32] also show that the global temperature increase using both RCP8.5 and SSP58.5 emission scenarios is virtually unchanged, rendering it likely that most of the additional warming we see in CMIP6 is due to updated model physics and subsequent changes in feedback strengths.

**Selection of GCMs/ESMs for downscaling**. For the downscaling of GCM data with our RCM MAR, we selected six CMIP5 and five CMIP6 models. The selection process of the CMIP5 models was done with the Ice Sheet Model Intercomparison Project for CMIP6 (ISMIP6) in mind and is fully described in[51]. The first step was to select a "Top 3" of models, which all fulfill a set of technical and statistical requirements: (i) output data must be available in 6 h intervals to force MAR at its lateral boundaries; (ii) 6 h output must be available for RCP2.6 and RCP8.5; (iii) compared to observations over the historical period, the essential statistical metrics must be above the median of the 33 model ensemble; and (iv) the essential climate metrics are not allowed to lie outside of two interquartile ranges of the median[51].

To form the "Top 6" ensemble for downscaling, the three additional models do not need to comply with the strict guidelines outlined by Barthel et al.[51], but they were chosen to maximize the projected diversity of twenty-first century climate change.

The bias and diversity of the Top 6 ensemble of all CMIP5 models were assessed by Barthel et al.[51] based on atmospheric and oceanic parameters. The resulting Top 3 models were HadGEM2-ES, MIROC5, and NorESM1-M, where MIROC5 was selected as the most representative for the median of the complete CMIP5 ensemble[51]. In addition, Barthel et al.[51] selected ACCESS1.3, CSIRO-Mk3-6-0, and IPSL-CM5A-MR, where CSIRO-Mk-3-6-0 projects a low atmospheric warming, ACCESS1.3 models geographically different patterns in warming and IPSL-CM5A-MR projects strong warming in the Greenland sea. Together, the selected Top 6 models should be representative of the mean and spread of the CMIP5 ensemble[51] and we chose the the r1i1p1 members of all six models.

Conversely, for the CMIP6 model selection, which are not part of the ISMIP6 project due to the delayed CMIP6 model data release, we were limited by the model availability. We wanted to follow a similar protocol as given in Barthel et al.[51], but after applying the first requirement, i.e. that MAR can only be forced by 6h outputs, we were only left with 5 out of 17 SSP58.5 models, of which we chose all 5. Our own analysis in this study (Fig. 1d) shows that these five randomly selected models already represent the ensemble mean, minima, and maxima of the full CMIP6 ensemble accurately. The models we chose for our analysis are—in alphabetical order—CESM2, CNRM-CM6-1, CNRM-ESM2-1, MRI-ESM2-0, and UKESM1-0-LL (validated over the current climate in Supplementary Figs. 1–5). From CESM2 and MRI-ESM2, we chose the r1i1p1f1 ensemble member, and from the two CNRM models and UKESM we chose r1i1p1f2.

We evaluated all our CMIP5 and CMIP6 MAR simulations (Supplementary Figs. 1–3) and the raw GCM output (Supplementary Figs. 4–5) for essential climate parameters during the 1981–2010 period. On average, MAR CMIP5 overestimates the SMB for each pixel by 26.3 mm w.e. (Supplementary Fig. 1), similar to MAR CMIP6 with a mean bias of 29.8 mm w.e. when compared to the reference simulation where we forced MAR with the ERA-Interim reanalysis[52]. Both model means underestimate the runoff during the 1981–2010 period, with MAR CMIP5 having a considerably higher bias of −21.2 mm w.e. compared to −13.0 in MAR CMIP6 (Supplementary Fig. 2). MAR CMIP5 produces a mean snowfall bias of 3.7 mm w.e., compared to 12.6 in MAR CMIP6 (Supplementary Fig. 3). Further, the chosen CMIP6 models have on average a lower bias of 2.6 gpm (geopotential meters at 500 hPa), compared to 23.2 gpm in the CMIP5 models (Supplementary Fig. 4) when compared to ERA-Interim. This lower bias of the 500 hPa geopotential indicates a better representation of the mid-atmospheric circulation in CMIP6 than in CMIP5 between 1981 and 2010. Further, the CMIP5 GCMs show a cold bias of −0.9 °C at 600 hPa compared to −1.9 °C at 600 hPa during summer in CMIP6 (Supplementary Fig. 5). Overall, the CMIP5 and CMIP6 simulations show a similar behavior during the historical period for the SMB, while the CMIP6 models appear to have improved their representation of the mid-tropospheric circulation over the GrIS.

**Modéle Atmosphérique Régional**. For the downscaling of coarse-resolution CMIP5 and CMIP6 data, we used the MAR, an open-source and widely used polar RCM[2,3,8,9,21,38–41,53–55]. MAR consists of a hydrostatic dynamical core which solves the primitive equation set[38,39]. A full description of the model setup, the underlying physical parameterizations and evaluation of MAR for polar climates are described in refs. [9,21,38–41,53,55]. In this study, we used the MARv3.9.6 version, evaluated in ref. [56], and the source code of MAR for the reproduction of this study is available via the MAR hompeage at http://mar.cnrs.fr.

Within MAR, the snow and ice properties at the ice sheet-atmosphere interface are calculated in the Soil Ice Vegetation Atmosphere Transfer module[38]. This module calculates the main snowpack based on the snow module CROCUS[57,58], but also handles the mass and energy exchange between the atmosphere (e.g. radiation, precipitation, temperature) and the bare-ice surfaces, the snowpack, and the Arctic tundra that surrounds the GrIS[38,40].

For the six CMIP5 and five CMIP6 future projections we downscaled, we prescribed the boundary conditions in exactly the same manner and also used the exact same MAR version and setup throughout. Overall, MAR was forced at its lateral boundaries (pressure, wind speed, temperature, specific humidity), at the top of the stratosphere (temperature, wind speed) and at the ocean surface (sea ice concentration, sea surface temperature) every 6 h using GCM and ERA-Interim reanalysis fields[2,40,53,55]. We ran MAR at a spatial resolution of 15 km x 15 km on a polar stereographic projection, which represents a significant increase in resolution compared to previous GrIS regional climate projections with MAR in[2], and which was used in the IPCC AR5[1]. The MAR setup used in this study has been thoroughly compared to observations from weather stations, observed radiative fluxes, satellite cloud cover, satellite albedo and melt extent, ablation, and SMB in situ measurements[21,40,41,56,59].

## Data availability

All the MAR model results are available for download on ftp://ftp.climato.be/fettweis/MARv3.9/ISMIP6/GrIS/ in the framework of the ISMIP6 exercise (https://tc.copernicus.org/articles/14/2331/2020/). CMIP5 and CMIP6 model outputs can be openly accessed via different ESGF data nodes (e.g., https://esgf-node.llnl.gov/projects/cmip6/, https://esgf-node.llnl.gov/projects/cmip5/).

## Code availability

All the code used for the analysis in this study is available upon request from the corresponding author.

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

## Acknowledgements

This project has received funding from the European Research Council (ERC) under the European Union's Horizon 2020 research and innovation program (Grant agreement number 758005) as well as from the Horizon 2020 PROTECT project no. 869304 (This is PROTECT contribution number 5). Computational resources have been provided by the Consortium des Equipements de Calcul Intensif (CECI), funded by the Fonds de la Recherche Scientifique de Belgique (F.R.S.-FNRS) under grant number 2.5020.11 and the Tier-1 supercomputer (Zenobe) of the Federation Wallonie Bruxelles infrastructure funded by the Walloon Region under the grant agreement number 1117545. This work was also supported by the Fonds de la Recherche Scientifique (FNRS) and the Fonds Wetenschappelijk Onderzoek-Vlaanderen (FWO) under the EOS Project number O0100718F. We thank Katherine Thayer-Calder and William Lipscomb for providing 6 hourly output data from CESM2.

## Author contributions

S.H., C.K., C.A., A.D., X.F., C.L., and A.T. designed the study. S.H. analyzed the data and wrote the manuscript. C.L. provided the analysis for the Supplementary Material. X.F. did the MAR simulations. All authors discussed the final version of manuscript.

## Competing interests

The authors declare no competing interests.
