## [Peer Review File · Nature Communications]

Reviewers' Comments:

Reviewer #2:

Remarks to the Author:

Hofer et al., Doubling of future Greenland Ice Sheet surface melt revealed by the new CMIP6 high emission scenario

Hofer et al present results for the Greenland ice sheet comparing the surface mass budget (precipitation, melt, runoff) from 6 CMIP5 models and 5 CMIP6 models, all downscaled to 15km with the MAR regional climate model. The downscaling allows the authors to estimate the likely consequences for Greenland ice sheet mass loss of two similar emissions scenarios, RCP8.5 from CMIP5 and SSP585 in CMIP6. Their results suggest that the CMIP6 models project more melt and ice loss from Greenland compared to the CMIP5 models.

This is a well-written and easy to follow paper with interesting results that I fully enjoyed reading. It will undoubtedly be an important contribution to the IPCC 6th assessment report as relatively few papers have been published so far considering the future of the Greenland ice sheet based on CMIP6 models. The results seem robust as far they go and are well explained. However, I feel that the conclusions they draw are somewhat overstated for two reasons:

Initially results from the first CMIP6 models released in the archive suggested a much higher equilibrium climate sensitivity than in CMIP5 that may explain the authors findings of increased warming and Arctic amplification in CMIP6. However, as more global models have come available it has become clear that the higher ECS is not seen in all models, thus the decision to use what was then available may bias the conclusion that CMIP6 has significantly more melt than CMIP5. This is particularly true as the CMIP5 models were chosen by the authors to span the width of the ensemble. It is not possible based on the data presented here to identify if this is the case with the CMIP6 models, which unfortunately slightly muddies the conclusions.

It is not totally clear how similar the RCP8.5 and SSP585 scenarios are through the whole 21st century. Although they end on the same Wm2 forcing, the evolution through the century is slightly different. This makes it hard to attribute the enhanced melt to either improved/changed model physics or differences in scenarios.

More generally, and given the wealth and potential of the data the authors have generated I felt the discussion was a little superficial. This is completely understandable given the time pressure of the CMIP6/ISMIP6 programmes but I hope the authors will use the opportunity presented by this review for some deeper analysis. While I appreciate there are space constraints, I felt the overall message was repeated too often and there are some scientifically and societally important outstanding questions that the authors have the opportunity to help to answer or at least point in the direction for future research. Some suggestions:

given that SSP585 is a very extreme (though not impossible) emissions scenario, how do these results help to provide the kind of guidance required by stakeholders as mentioned in the conclusion section, when considering lower scenarios?

Is there a simple scaling between temperature and melt as implied by the authors or do more complex factors play a role?

The authors mention but do not explicitly examine the effects of the ability of the models to characterise atmospheric circulation patterns that strongly influence melt rates and precipitation amounts. Is downwelling short wave an important contribution to enhanced melt?

Clouds are well known from the authors previous work to affect ice sheet melt rates but are not considered in any significant detail here where the focus is only on temperature.

More specifically. I would like to see some discussion from the authors distinguishing between changes in the driving models from CMIP5/6 and differences introduced by the different scenarios RCP8.5/SSP585

Plotting the two scenarios and the modelled SMB from the two ensembles as a time varying variable through the 21st century would assist here. In general, more detail on the actual SMB numbers from each model (perhaps in a table) would also assist the reader in being able to tease out how different the numbers for each model actually are from each other. Does excluding one or other model from the two ensembles significantly change the conclusions?

Some more minor comments:

Page 3. "The latest GCM model suite is now more sensitive to atmospheric greenhouse gases"

As mentioned above, a bigger ensemble now available suggests that this is not true of all models, though it would be interesting to examine if those with little change in ECS are those that are essentially unchanged, or if other factors can explain it.

Given recent controversies in the literature it may be worth inserting a sentence to emphasise that RCP8.5/SSP585 are extreme greenhouse gas emissions scenarios.

Page 4.

"globally CMIP5 and CMIP6 projections start to diverge around 2050 but already in 2030 in the Arctic" I assume that here it is surface temperature that diverges, but I suppose it could be emissions? Please clarify this and as suggested above I would like to see the emissions from the two scenarios mapped over the temperature plots to see how they align. I would imagine some of the divergence is attributable to differences in timing of sea ice disappearance, it would be interesting to compare this too.

Page 7.

The differences in melt and runoff between the two ensembles is the whole point of the paper, but there isn't much comparison of changes in downwelling short wave or clouds. This should at least be acknowledged as an important source of melt even if there is insufficient space for further figures.

Page 8.

"One potential explanation is the higher ECS in CMIP6 models"

I agree this is quite likely as an explanation but there are other plausible explanations or contributing factors including improved clouds/humidity, circulation variability, improved sea ice modelling giving steeper Arctic amplification and these should at least be mentioned.

page 9. Figure 3 is very interesting but I'd like to see a figure that gives annual anomalies as well as cumulative anomalies. These are more easily compared with emissions scenario curves and potentially other variables such as declining sea ice.

page 11 Figure 4 shows convincingly the enhanced arctic amplification in the CMIP6 models but looking at 4.b I wonder if all of the models show the 8-10K increase in JJA 600 hPa temperature in CMIP6 or if it is possible that only one or two do? These kind of relatively small ensembles can easily give skewed results when only a few models are included (though in fact 5 or 6 models is actually quite a large number when applied to the ice sheet compared to previous studies and the authors are to be commended for taking this on!). In any case, this is important to clarify given the conclusions drawn on page 12.

Page 12/13. It's not really explicitly mentioned in the paper but the historical scenario models for CMIP5 typically underestimate melt from the Greenland ice sheet, is this true of CMIP6? In which case what are the implications for other scenarios? The authors mention the inability of CMIP5 models to reproduce Arctic circulation anomalies but it's not clear if this plays a role in the enhanced melting or if in fact it is a possible source of even greater melt.

Paragraph 2 on page 13 is important justification for this work and some general assessment of how other emissions scenarios will be affected would be helpful, though maybe beyond the scope of the paper. Nonetheless I think it should at least be mentioned.

Page 19: Describing RCP8.5 as "business as usual" is a bit risky as our current trajectory is not RCP8.5, I would suggest changing this to "high emissions scenario":

Page 20/21. As mentioned above a figure showing the actual emissions curves for the two scenarios and perhaps how they relate to other scenarios would be helpful context here.

Page 22: It's clear from the models you chose that they are ESMs and as far as I know all have multi-member ensembles, which ensemble members did you choose (is that what the final digit refers to?) and how similar are the other ensemble members to the ones chosen? I appreciate this information may not be available but I suspect that e.g. CESM2 has had considerable work published on the ensemble and this is probably more generally applicable and will help to assess the uncertainty within the models. Some discussion on this seems important.

page 23:

Typo on line 119

Lines 115 to 117 on geopotential height are a bit of a non-sequitur - what is the significance of this bias reduction?

Line 125 - is MAR hydrostatic or non-hydrostatic?

Page 26:

What does the hatching indicate in Figure S1 etc..?

It would be helpful to have the mean annual SMB figures for each model to understand what the actual range is. These numbers would also be useful for e.g. a 30 year timeslice at the end of the 21st century. I suggest releasing the actual absolute integrated SMB over the ice sheet in order to assess how these models compare with other models as they are released.

Reviewer #3:

Remarks to the Author:

Review of Doubling of future Greenland Ice Sheet surface melt revealed by the new CMIP6 high-emission scenario

I think this is a nice first study looking at differences in Greenland surface melting between CMIP5 and CMIP6 projections. It follows earlier studies that used CMIP5 output within MAR to forecast changes in Greenland SMB. I'm not sure it's a significant advancement in understanding over earlier CMIP5 papers and thus not sure if it's really a Nature paper, but that's something for the editor to decide.

I did feel that it left some important details out, especially in regards the seasonality of the temperature differences between CMIP5 and CMIP6 which are important to address. And I would like to see more discussion on how the temperature differences by the end of the century could increase melt that much, there must be something else going on, unless the number of melting degree days can actually lead to that much extra melt. I would also like to see spatially where the increase melt happens, and I would also like to see how the winter accumulation compares between CMIP5/6. Arctic amplification is mostly an autumn/winter phenomena related to the ice-albedo feedback and in spring with changes in atmospheric circulation patterns that advect warm/moist air into the Arctic but in summer there hasn't been much of a temperature increase. Thus, saying there is enhanced Arctic amplification without really looking into the seasonality of this is somewhat misleading. Also Arctic amplification has to do with outsized warming in the Arctic compared to mid-latitudes, and it's unclear if that proportion has changed, or if it's just that there is more warming overall in CMIP6, which I believe is the case. With some more analysis and clarification I think this will be a nice contribution.

1. It may be good in the introduction to briefly mention that surface melt dominates Greenland mass loss today whereas in the past it was balanced by melt and discharge and how other studies have assumed this dominance of surface melt to mass balance loss will continue through the end of the century.

2. I am a bit concerned about the greater temperature rise in the Arctic in CMIP6 than in CMIP5. I guess these are annual values, but more interesting in terms of Greenland mass balance is the seasonality of these temperature difference. That is needed in order to really understand the doubling of surface melting during the 21st century.

3. In the introduction I think it would be good to also discuss why you do the downscaling and run MAR instead of use the output directly from the CMIP5/6 models to assess changes in surface melt.

4. At first reading the main body I didn't understand why you didn't use all the models for Figure 2 and instead used a subset. It's explained in the supplemental material but maybe just mention in the main body that you were limited by the number of models that supplied 6 hourly fields needed by MAR.

5. It is curious why the mass loss deviates already in 2020 as the temperature differences happen a few decades later. This needs some discussion and perhaps it relates to seasonal temperature differences which is something you are not showing here. The timing of when the melt season starts and ends is important here and I feel this should form part of your investigation. You do mention later that there are differences in the ablation season but you don't show the temperatures as a function of month which I think would be helpful here to see how those differ between CMIP5/6.

6. I don't understand the statement on page 8 "The CMIP6 forced MAR simulations therefore do not have an inherent bias in SMB". How have you shown this? You end that paragraph with a comment on the ECS but you don't explain how that could explain what you have discussed in that paragraph. I feel this entire section needs to be expanded upon.

7. On page 10 you say the melt season extends, but this 22 days, is that by the end of the century? That doesn't really seem like a large extension in the melt season. In the Arctic Ocean places already have extended melt season of more than 6 weeks between 1980 and today.

8. And are you also saying on page 10 that there are no differences in winter accumulation between CMIP5/6? And that winter temperatures are similar? Arctic amplification tends to be most prominent in autumn and winter, not in summer, so this is counter to what you are suggesting, in that you seem to suggest summer temperatures are the big difference between CMIP5/6, which is not really an Arctic amplification signal related to any ice-albedo feedbacks, but rather changes in atmospheric circulation, clouds and water vapor.

9. In your discussion, since you are talking about Arctic air temperatures, the longer melt season at the end of the melt is more because of the loss of sea ice and increased ocean mixed layer temperatures which delays freeze-up of the ocean and therefore keeps Arctic temperatures warmer longer. I realize you talk about changes in the melt season for the whole of Greenland but be good to see regionally where this is as there may be links to sea ice cover.

REVIEWER COMMENTS

Black font = Reviewer comments

Blue font = Answers to comments

Red font = Changes in the text of the manuscript

Reviewer #2 (Remarks to the Author):

We would like to thank Reviewer #2 for their thorough assessment of our manuscript and we hope to have addressed most of the concerns in our answers below. We think the paper does now rest on a more solid foundation.

Hofer et al., Doubling of future Greenland Ice Sheet surface melt revealed by the new CMIP6 high emission scenario

Hofer et al present results for the Greenland ice sheet comparing the surface mass budget (precipitation, melt, runoff) from 6 CMIP5 models and 5 CMIP6 models, all downscaled to 15km with the MAR regional climate model. The downscaling allows the authors to estimate the likely consequences for Greenland ice sheet mass loss of two similar emissions scenarios, RCP8.5 from CMIP5 and SSP585 in CMIP6. Their results suggest that the CMIP6 models project more melt and ice loss from Greenland compared to the CMIP5 models.

This is a well-written and easy to follow paper with interesting results that I fully enjoyed reading. It will undoubtedly be an important contribution to the IPCC 6th assessment report as relatively few papers have been published so far considering the future of the Greenland ice sheet based on CMIP6 models. The results seem robust as far they go and are well explained. However, I feel that the conclusions they draw are somewhat overstated for two reasons:

Initially results from the first CMIP6 models released in the archive suggested a much higher equilibrium climate sensitivity than in CMIP5 that may explain the authors findings of increased warming and Arctic amplification in CMIP6. However, as more global models have come available it has become clear that the higher ECS is not seen in all models, thus the decision to use what was then available may bias the conclusion that CMIP6 has significantly more melt than CMIP5. This is particularly true as the CMIP5 models were chosen by the authors to span the width of the ensemble. It is not possible based on the data presented here to identify if this is the case with the CMIP6 models, which unfortunately slightly muddies the conclusions.

We think Reviewer #2 has made a really interesting point here, regarding changes in our CMIP6 ensemble and representativeness of our chosen subset for downscaling, due to the fact that the CMIP6 ensemble was still evolving when we submitted the manuscript. Since receiving the manuscript back, we have now included all CMIP6 models that have recently become available and we now consider equally-sized ensembles of 28 CMIP5 and 28 CMIP6 (RCP8.5 and SSP58.5) models. This inclusion of additional models in CMIP6 means that we now consider 11 more CMIP6 models. These 11 models also include the models which Reviewer #2 mentions to have a lower equilibrium climate sensitivity (ECS), which can also be seen in our Figure 1, where the difference in Arctic amplification between CMIP6 and CMIP5 has been slightly reduced from +1.7C to +1.3C.

While the surplus in 21st century warming has been slightly reduced, the CMIP6 SSP58.5 ensembles still produces a +1.3C greater Arctic warming, which is now based on an equal sample size as the CMIP5 RCP8.5 ensemble (28 models each). Therefore, we think that our conclusion that CMIP6 warms more than CMIP5 – especially in the Arctic - for the same greenhouse gas forcing is

robust, and also that our chosen sample for downscaling is still representative of the overall CMIP5 and CMIP6 means.

It is not totally clear how similar the RCP8.5 and SSP585 scenarios are through the whole 21st century. Although they end on the same Wm^2 forcing, the evolution through the century is slightly different. This makes it hard to attribute the enhanced melt to either improved/changed model physics or differences in scenarios.

More generally, and given the wealth and potential of the data the authors have generated I felt the discussion was a little superficial. This is completely understandable given the time pressure of the CMIP6/ISMIP6 programmes but I hope the authors will use the opportunity presented by this review for some deeper analysis. While I appreciate there are space constraints, I felt the overall message was repeated too often and there are some scientifically and societally important outstanding questions that the authors have the opportunity to help to answer or at least point in the direction for future research. Some suggestions:

given that SSP585 is a very extreme (though not impossible) emissions scenario, how do these results help to provide the kind of guidance required by stakeholders as mentioned in the conclusion section, when considering lower scenarios?

Is there a simple scaling between temperature and melt as implied by the authors or do more complex factors play a role?

Regarding the simple scaling and how this study can be a guidance for stakeholders using lower emission scenarios: we think that this is an excellent point and have therefore included one new subplot (Fig.4C) that shows the GrIS temperature anomaly from GCMs vs. the GrIS annual SMB in MAR. Further, we have included a completely new set of figures, namely Figure 5 – showing the 21st century evolution of GrIS SMB across low-, mid- and extreme emission scenarios (RCP2.6, RCP4.5, RCP8.5, SSP126, SSP245, SSP585) reconstructed for all available GCMs using the equation in Figure 4C. Figure 5, together with the new Table 1 – showing the mean and standard deviations of the Greenland sea level rise contribution and time when the SMB drops below 0 Gt/yr – will hopefully also be of value to stakeholders mentioned by Reviewer #2. We have also included multiple paragraphs throughout the manuscript discussing these new results. We think that our approach – using 11 MAR simulations (6 RCP8.5, 5 SSP585) - allows us a more robust statistical reconstruction of the SMB across other emission scenarios, given that in the literature these reconstructions are often based on a single GCM input model. However, our results clearly show that there is some variability between individual model realisations, and picking one individual correlation for reconstruction might lead to significant biases (e.g. V.d.Broeke (2020)).

To discuss the differences we find between the CMIP6 and CMIP5 scenarios, we are now also discussing more complex factors. Figure 7 shows the difference in the main surface energy budget components, leading to the conclusion that the reduction in SWD with increases in LWD over similar areas are due to cloud optical depth and longwave emissivity feedbacks. Additionally, we also find that the main differences in radiative fluxes between our 5 SSP585 and 6 RCP8.5 MAR simulations are focused on the summer melt season, and that the main difference is still a greater absorbed SW flux at the surface. Further, we have also included a new figure (Fig.8) showing the mean 2071-2100 sea ice extent around Greenland. We think that some of the greater cloud optical depth might be due to more open sea water and greater moisture availability around Greenland. However, we cannot assess how much of the greater COD is due to conversion of ice clouds to mixed-phase or liquid clouds, or due to large-scale advection of humidity on the synoptic scale. All of these mentioned new figures are discussed throughout the manuscript.

The authors mention but do not explicitly examine the effects of the ability of the models to characterise atmospheric circulation patterns that strongly influence melt rates and precipitation amounts. Is downwelling short wave an important contribution to enhanced melt?

Clouds are well known from the authors previous work to affect ice sheet melt rates but are not considered in any significant detail here where the focus is only on temperature.

More specifically. I would like to see some discussion from the authors distinguishing between changes in the driving models from CMIP5/6 and differences introduced by the different scenarios RCP8.5/SSP585

Reviewer #2 raises some interesting concerns here. Re. 1) How different are RCP8.5 and SSP58.5 and how much of the increased warming in CMIP6 is due to changes in model physics and how much is due to differences in scenarios. In short, the SSP scenarios have been designed to provide a new set of scenarios, that should be directly comparable to the RCP radiative forcing levels, despite changes in timing of emissions of different climate forcers (Riahi et al. 2017). For example, while in SSP58.5 the CO₂ emissions are higher than in RCP8.5, the opposite is true for a different potent greenhouse gas, namely CH₄ (Riahi et al. 2017). Therefore, while the forcing might come from different sources, the scenarios should by design be comparable to each other (see also the Figure below from O'Neill et al. (2016) (similar to the analysis of Riahi et al. (2017)).

The figure below from O'Neill et al. (2016) shows that the global mean temperature change and radiative forcing in SSP58.5 and RCP8.5 are virtually the same until 2300 when using the MAGICC model with unchanged physics throughout. At the end of the 21st century the difference in global temperature lies around 0.1C using an unchanged climate model, despite different CO₂ and fossil fuel emissions (first row). Therefore, it is likely that the underlying cause in temperature rise is due to updated physics in individual CMIP6 models, and not due to differences in the forcing scenario. We have highlighted this interpretation in the Methods of our manuscript:

P31 L531-538: “While this is an interesting result, the fact that such an analysis is yet to be done for most other CMIP6 ensemble members means that the research on the underlying causes of the greater 21st temperature increase in CMIP6 compared to CMIP5 for the same radiative forcing is still emerging. However, using an unchanged climate model (MAGICC version 6.8.01), O’Neill et al. [30] also show that the global temperature increase using both RCP8.5 and SSP58.5 emission scenarios is virtually unchanged, rendering it likely that most of the additional warming we see in CMIP6 is due to updated model physics and subsequent changes in feedback strengths.

Regarding the questions dealing with “are shortwave/clouds/circulation anomalies an important factor?”. Both sets of ensembles – CMIP5 and CMIP6 – don’t reproduce the recent anticyclonic circulation anomalies, which is one driver of potential uncertainty in future simulations as discussed in Delhasse et al. (2018) and in Hofer et al. (2019). Reviewer #2 is right that we didn’t discuss this in the earlier version of the manuscript, and therefore we have now included a new figure showing the difference in GrIS surface energy budget between CMIP6 SSP585 and CMIP5 RCP8.5 MAR simulations, and we have also included a new figure showing the difference in sea ice extent between our chosen subset of CMIP5 and CMIP6 GCMs.

Plotting the two scenarios and the modelled SMB from the two ensembles as a time varying variable through the 21st century would assist here. In general, more detail on the actual SMB numbers from each model (perhaps in a table) would also assist the reader in being able to tease out how different the numbers for each model actually are from each other. Does excluding one or other model from the two ensembles significantly change the conclusions?

As discussed in a previous comment, we now show the reconstructed SMB for all CMIP5 and CMIP6 scenarios in Figure 5, together with a table indicating the exact Greenland sea level rise contribution throughout the 21st century, together with upper and lower statistical bounds. Additionally, all the MAR model results are available for download on <ftp://ftp.climato.be/fettweis/MARv3.9/ISMIP6/GrIS/> in the framework of the ISMIP6 exercise (<https://tc.copernicus.org/articles/14/2331/2020/>), including the SMB for each individual model. We think it is more robust for the user to download the data and use their own ice sheet mask, because the Greenland Ice Sheet SMB will depend on the individual model's mask to allow exact cross-comparisons.

Additionally, excluding individual models from our results doesn't significantly change our results. For example, in Figure 3B, the cumulative SMB anomalies in 2100 from RCP8.5 are all less than the SSP585 MAR SMB anomalies. The MAR SSP585 run with the least amount of surface mass loss throughout the 21st century has still more mass loss than the RCP8.5 simulation with the most mass loss. Of course the exact numbers would vary to a few percent, but the bigger picture and storyline would stay intact, even when excluding one member of the MAR ensemble. This can also now be seen from the fact that our sea level rise estimates lie close to the mean of the reconstructed values in Table 1 for the overall CMIP ensembles.

Some more minor comments:

Page 3. "The latest GCM model suite is now more sensitive to atmospheric greenhouse gases"

As mentioned above, a bigger ensemble now available suggests that this is not true of all models, though it would be interesting to examine if those with little change in ECS are those that are essentially unchanged, or if other factors can explain it.

Given recent controversies in the literature it may be worth inserting a sentence to emphasise that RCP8.5/SSP585 are extreme greenhouse gas emissions scenarios.

We now consider a similar-sized CMIP6 ensemble (28 members) and Figure 1 still clearly shows that CMIP6 SSP58.5 warms more and faster than CMIP5 RCP8.5. We therefore consider our chosen analysis to be based on a solid foundation.

We agree with the Reviewer's comment to mention that these scenarios are "extreme" members of the CMIP family, and we have therefore changed the following sentences:

P3 L40 "Here we show that between the high-emission scenario from CMIP5 (RCP8.5) and CMIP6 (SSP58.5) [29, 30, 33], which share a similar **extreme** surface warming of 8.5 W/m² in 2100, GrIS surface melting almost doubles during the 21st century."

We also changed "business as usual scenario" to "extreme high-emission scenario" following a comment by Reviewer #2 below, in addition to highlighting that RCP8.5 and SSP585 are "extreme high-emission scenarios" throughout the manuscript.

Page 4.

"globally CMIP5 and CMIP6 projections start to diverge around 2050 but already in 2030 in the Arctic" I assume that here it is surface temperature that diverges, but I suppose it could be emissions? Please clarify this and as suggested above I would like to see the emissions from the two scenarios mapped over the temperature plots to see how they align. I would imagine some of the divergence is attributable to differences in timing of sea ice disappearance, it would be interesting to compare this too.

Agreed. Regarding the sea ice comparison, we have now included a new plot showing the mean sea ice extent around Greenland in CMIP5 and CMIP6 at the end of the 21st century. This might be indicative of why there are a notably lower downwelling shortwave fluxes and greater longwave downward fluxes (see new Fig.8), i.e. a greater moisture availability in CMIP6 due to more open ocean leading to a higher cloud optical depth (COD) and longwave emissivity as discussed in Hofer et al. (2019). However, we cannot assess how much of the increase in COD is due to a phase change within existing clouds from ice to mixed-phase or liquid or how much moisture is advected from remote areas, which will have to be discussed in follow-up manuscripts.

We have added the following paragraph to the main part of the manuscript: **P19-21 L237-L254** "One of the potential hypotheses for a greater cloud optical depth during summer melt season towards the end of the 21st century is a greater loss of sea ice in the 5 CMIP6 models we chose for downscaling (Fig.6). The mean summer location of the southern edge of the sea ice at the end of the 21st century (2071-2100) in our 5 CMIP6 models in the Baffin Bay sector migrates approximately 1000 km further north along Baffin Island and moves roughly 400 km north in the northwest of Greenland compared to our 6 CMIP5 models (Fig.6). Conversely, in the Greenland Sea sector east of Greenland the southern sea ice edge lies roughly 600 km north where it is connected to Greenland, but less so over the open ocean. Generally, in our CMIP6 models only the northernmost edge of Greenland, where most of the multi-year sea ice resides, is still covered in sea ice between 2070-2100. Apart from the north, Greenland will be surrounded by open water and a quasi-unlimited moisture and heat source in CMIP6 at the end of the 21st century during summer. While this likely explains part of the increase in cloud optical depth seen from the surface energy budget analysis in Fig.5, we cannot assess from our data how

much of the increase in cloud optical thickness is due to a phase change from mostly ice-containing clouds to mixed-phase or liquid clouds [7, 14, 15]. Additionally, recent studies using lagrangian moisture tracking also point towards that a notable proportion of humidity is advected to the GrIS on the synoptic-scale and not locally from the surrounding ocean [44], which will have to be evaluated in more detail in future studies.

Regarding the comparison between (local) emissions and the Arctic temperature time series: We don't think it is feasible for us to discuss the top-of-the-atmosphere energy fluxes in the GCMs, the emissions and the temperature from all GCM models within the scope of this study. While this is an interesting point, this will have to be discussed in a different set of publications.

Page 7.

The differences in melt and runoff between the two ensembles is the whole point of the paper, but there isn't much comparison of changes in downwelling short wave or clouds. This should at least be acknowledged as an important source of melt even if there is insufficient space for further figures.

This manuscript was initially written for a different journal within the same group of journals which has a much more concise format. Due to additional space and this generally being a very good point by Reviewer #2, we have included a new figure (Fig.7) showing the surface energy budget difference between our MAR CMIP6 (SSP585) and CMIP5 (RCP8.5) simulations for 2071-2100 for all seasons (SWD, Swnet and LWD fluxes). We have also added new paragraphs discussing this figure and cloud optical depth feedbacks (partly included in a previous comment).

Page 8.

“One potential explanation is the higher ECS in CMIP6 models”

I agree this is quite likely as an explanation but there are other plausible explanations or contributing factors including improved clouds/humidity, circulation variability, improved sea ice modelling giving steeper Arctic amplification and these should at least be mentioned.

Please refer to previous answers for a discussion of these valid points, but we have included further results and discussions on the influence of the Greenland Ice Sheet circulation, surface energy budget, clouds and the sea ice in our models used for downscaling. We hope that this addresses Reviewer #2's comment.

page 9. Figure 3 is very interesting but I'd like to see a figure that gives annual anomalies as well as cumulative anomalies. These are more easily compared with emissions scenario curves and potentially other variables such as declining sea ice.

We present a new figure showing the annual SMB across all CMIP5 and CMIP6 emission scenarios for 1960-2100, and a new table that presents exact numbers for cumulative anomalies (and sea level rise equivalent) and statistical upper and lower bounds. All the MAR model results are available for download on <ftp://ftp.climato.be/fettweis/MARv3.9/ISMIP6/GrIS/> in the framework of the ISMIP6 exercise (<https://tc.copernicus.org/articles/14/2331/2020/>), for which all the SMB numbers for a given ice sheet mask can be extracted.

page 11 Figure 4 shows convincingly the enhanced arctic amplification in the CMIP6 models but looking at 4.b I wonder if all of the models show the 8-10K increase in JJA 600 hPa temperature in CMIP6 or if it is possible that only one or two do? These kind of relatively small ensembles can easily give skewed results when only a few models are included (though in fact 5 or 6 models is actually quite a large number when applied to the ice sheet compared to previous studies and the authors are to be commended for taking this on!). In any case, this is important to clarify given the conclusions drawn on **page 12**.

We have tried to randomly remove various different ensemble members from this figure and redo the 2nd order polynomial fit, without any major changes to the equation of the fit or the correlation score. Therefore, we have not included any new analysis based on this comment. Further, we think that our chosen subset of CMIP6 SSP585 models is robust, given the analysis presented in Figure 1 regarding the global and Arctic warming, and Figure 5 and Table 1, both showing that our RCM-based estimates and the reconstructed SMB for the overall ensemble don't differ notably.

Page 12/13. It's not really explicitly mentioned in the paper but the historical scenario models for CMIP5 typically underestimate melt from the Greenland ice sheet, is this true of CMIP6? In which case what are the implications for other scenarios? The authors mention the inability of CMIP5 models to reproduce Arctic circulation anomalies but it's not clear if this plays a role in the enhanced melting or if in fact it is a possible source of even greater melt.

We have updated our discussion to now include that missing anticyclonic circulation anomalies could be a source of even greater GrIS melt until 2100:

P23 L286 “Conversely, CMIP6-based melt estimates presented in our study could also be too low, given that both CMIP5 and CMIP6 models currently lack the ability to accurately reproduce recent Arctic general circulation anomalies, which have been responsible for recent record mass losses and could be a source of greater 21 st century GrIS mass loss [7, 8, 19].”

Additionally, the evaluation of MAR SSP585 and RCP8.5 over the current climate (Fig. S1-S5) produces no significant differences in SMB or melt over the present-day. But it is true that both, MAR RCP8.5 and SSP585, slightly overestimate present-day SMB, but MAR CMIP6 has a considerably lower mean bias in surface runoff over the current climate (-21.2 mm w.e. vs -13.0 mm w.e., see methods). Based on this comment and one by Reviewer #3 we have included the following sentence in the main part of the manuscript to highlight that there are no significant differences in SMB when compared to MAR forced by observations:

P10 L116-120 “Fig. S1-S5 show that over the current climate the SMB and other variables in MAR forced by reanalysis compared to GCM forcing do not differ significantly. Over the current climate (1981-2010) the SMB in MAR forced by SSP585 models does not show an inherent bias when compared to the MAR RCP8.5 simulations (mean bias: 29.8 mm w.e. (SSP585) compared to 26.3 mm w.e. (RCP8.5)).”

Paragraph 2 on page 13 is important justification for this work and some general assessment of how other emissions scenarios will be affected would be helpful, though maybe beyond the scope of the paper. Nonetheless I think it should at least be mentioned.

We fully agree with this. Therefore, we now also reconstruct the SMB across all CMIP5 and CMIP6 scenarios. Please see other comments for a longer discussion on this matter.

Page 19: Describing RCP8.5 as “business as usual” is a bit risky as our current trajectory is not RCP8.5, I would suggest changing this to “high emissions scenario”

Thank you, that is a very good observation! We followed the recent discussion about the plausibility and naming of extreme end-members of the emissions scenarios and therefore agree that business-as-usual is not a good term to use. We changed it to “For our downscaled CMIP5 future simulations we use the **extreme high-emission scenario RCP8.5 (2006-2100)**” **P30 L494**

Page 20/21. As mentioned above a figure showing the actual emissions curves for the two scenarios and perhaps how they relate to other scenarios would be helpful context here. Please see earlier discussion on this point.

Page 22: It's clear from the models you chose that they are ESMs and as far as I know all have multi-member ensembles, which ensemble members did you choose (is that what the final digit refers to?) and how similar are the other ensemble members to the ones chosen? I appreciate this information may not be available but I suspect that e.g. CESM2 has had considerable work published on the ensemble and this is probably more generally applicable and will help to assess the uncertainty within the models. Some discussion on this seems important.

For the CMIP5 models used for downscaling with MAR we exclusively used the “r1i1p1” ensemble members, which essentially boils down to the first *realisation*, first *initialisation* and first set of *physics*. For our CMIP6 subset we used the “r1i1p1f1” ensemble members for two models (CESM2, MRI-ESM2), while we used the “r1i1p1f2” members with *forcing index* 2 for both CNRM models and the UKESM.

We included the following information in the main text of the manuscript to identify the ensemble members we used:

P8 L70 -73: “We chose 6 CMIP5 GCMs (HadGEM2-ES, MIROC5, NorESM1-M, ACCESS1.3, CSIRO-Mk3-6-0, IPSL-CM5A-MR - **all r1i1p1 ensemble members**) and 5 CMIP6 GCMs (CESM2, CNRM-CM6-1, CNRM-ESM2-1, MRI-ESM2-0 and UKESM1-0-LL - **r1i1p1f1 ensemble members for CESM2 and MRI-ESM2, but r1i1p1f2 for CNRM and UKESM**) for our dynamical downscaling with MAR.”

We also included the following information in the Methods section where we discuss the selection of models for downscaling:

P33 L571-573: “Together the selected Top 6 models should be representative of the mean and spread of the CMIP5 ensemble [42] and we chose the the r1i1p1 members of all 6 models.“ and

P33 L582: “From CESM2 and MRI-ESM2 we chose the r1i1p1f1 ensemble member, and from the two CNRM models and UKESM we chose r1i1p1f2.”

Additionally, we agree with Reviewer #2 about the general interest of discussing the ensemble spread within one model. However, this proves to be almost impossible to do at this stage due to notable restrictions on the availability of CMIP6 model data on the ESGF data server. At this stage, we struggled for example to get the all the outputs of only *one* member of the ensemble from the two CNRM models for example to assess the sea ice distribution during melt season in the newly added Figure 8. Therefore, we were unfortunately not able to assess the spread within the model ensembles themselves in this publication.

page 23:

Typo on line 119 Changed “CMIP5 and CMIP5” to “CMIP5 and CMIP6”

Lines 115 to 117 on geopotential height are a bit of a non-sequitur - what is the significance of this bias reduction? We have now included a sentence stating that “This lower bias of the 500 hPa geopotential indicates a better representation of the mid-atmospheric circulation in CMIP6 than in CMIP5 between 1981 and 2010.” **P34 L592-595**

Line 125 - is MAR hydrostatic or non-hydrostatic?

The operational MAR model used in this study is hydrostatic (“MAR consists of a hydrostatic dynamical core which solves the primitive equation set” P35 L603).

Page 26:

What does the hatching indicate in Figure S1 etc..? This is a good point. We have updated the figure captions in the supplements to now also include what the hatching means.

P38 Fig. S1: “The hatching in the figure indicates pixels where the absolute difference between MAR-GCM minus MAR-ERA-Int is lower than the interannual variability (standard deviation) for a given pixel.”

It would be helpful to have the mean annual SMB figures for each model to understand what the actual range is. These numbers would also be useful for e.g. a 30 year timeslice at the end of the 21st century. I suggest releasing the actual absolute integrated SMB over the ice sheet in order to assess how these models compare with other models as they are released. We have implemented a new figure showing the mean annual SMB across all emission scenarios as a time series throughout the 21st century. Additionally, all our MAR model results are available for download on <ftp://ftp.climato.be/fettweis/MARv3.9/ISMIP6/GrIS/> in the framework of the ISMIP6 exercise (<https://tc.copernicus.org/articles/14/2331/2020/>). However, for a direct comparison with other regional climate models one would have to make sure to use the same ice sheet mask, therefore we have not included the specific SMB values requested by the Reviewer here, but would like to refer to the underlying data for a more accurate individual comparison with other polar regional climate models.

References

van den Broeke, M., Noël, B., van Kampenhout, L., and van de Berg, W.-J.: A regional atmospheric warming threshold for irreversible Greenland ice sheet mass loss, EGU General Assembly 2020, Online, 4–8 May 2020, EGU2020-19032, <https://doi.org/10.5194/egusphere-egu2020-19032>, 2020

Reviewer #3 (Remarks to the Author):

Review of Doubling of future Greenland Ice Sheet surface melt revealed by the new CMIP6 high-emission scenario

I think this is a nice first study looking at differences in Greenland surface melting between CMIP5 and CMIP6 projections. It follows earlier studies that used CMIP5 output within MAR to forecast changes in Greenland SMB. I'm not sure it's a significant advancement in understanding over earlier CMIP5 papers and thus not sure if it's really a Nature paper, but that's something for the editor to decide.

I did feel that it left some important details out, especially in regards the seasonality of the temperature differences between CMIP5 and CMIP6 which are important to address. And I would like to see more discussion on how the temperature differences by the end of the century could increase melt that much, there must be something else going on, unless the number of melting degree days can actually lead to that much extra melt. I would also like to see spatially where the increase melt happens, and I would also like to see how the winter accumulation compares between CMIP5/6. Arctic amplification is mostly an autumn/winter phenomena related to the ice-albedo feedback and in spring with changes in atmospheric circulation patterns that advect warm/moist air into the Arctic but in summer there hasn't been much of a temperature increase. Thus, saying there is enhanced Arctic amplification without really looking into the seasonality of this is somewhat misleading. Also Arctic amplification has to do with outsized warming in the Arctic compared to mid-latitudes, and it's unclear if that proportion has changed, or if it's just that there is more warming overall in CMIP6, which I believe is the case. With some more analysis and clarification I think this will be a nice contribution.

First, we would like to thank Reviewer #3 for taking the time to provide valuable comments, which certainly have improved our manuscript. We hope to have addressed most of the concerns in the answers below.

Regarding the concerns outlined in the first two paragraphs

(1) Leaving out the seasonal aspect of Arctic temperature rise and whether the signal in the data is really an Arctic amplification signal.

Based on this valid concern we have included a new graphic (Fig.2) and text in the manuscript discussing these new results. Figure 2 shows the temperature increase for each season individually between 90S and 90N, for CMIP5 RCP8.5 and CMIP5 SSP585 and the difference between the two. Three things stand out. First, the warming in the Arctic is disproportional to the rest of the Earth, more than +10C, while the global average lies around 5-6C. Second, as pointed out by the reviewer, the data shows a strong winter peak in Arctic warming, of +16.9 C (2071-2100) in CMIP6, and two degrees less in CMIP5 RCP8.5, and an overall notable seasonal signal. Third, also the difference between the RCP8.5 and SSP585 shows a seasonal signal, leading us to the conclusion that what we see in the data is indeed an Arctic amplification signal and not a spatially and temporally uniform warming.

(2) Factors other than temperature - such as sea-ice feedbacks – playing a role:

We have included new figures showing the sea ice extent of the continuous sea ice at the end of the century in our RCP8.5 and SSP585 subset to highlight the difference in moisture and heat sources around Greenland between CMIP5 and CMIP6. In SSP585 there is significantly more open ocean than in RCP8.5 at the end of the century. Additionally, we also looked in the spatial patterns of the differences in the surface energy budget for each season in our high-resolution regional climate model simulations. We found that basically all of the differences in radiative fluxes at the surface are concentrated during summer melt season. Further, we think that there is potentially also a cloud optical depth feedback at play – regions show less downwelling shortwave fluxes but increased downwelling longwave fluxes at the same time during summer – as discussed in Hofer et al. (2019). We think that both of these factors -sea ice and clouds – might be connected to a degree, which is what we also try to discuss in the updated text of the manuscript.

Regarding also including the winter accumulation differences between the MAR simulations from SSP585 and RCP8.5: Our Figure 6 A) and B) show the cumulative anomalies of the annual accumulation terms (rainfall and snowfall). We looked into the differences during winter, and found no significant differences, as also highlighted by the annual surface mass balance cycle in Figure 6C). We therefore have not included any new analysis discussing the differences in winter accumulation explicitly to save space for other new results and discussions.

1. It may be good in the introduction to briefly mention that surface melt dominates Greenland mass loss today whereas in the past it was balanced by melt and discharge and how other studies have assumed this dominance of surface melt to mass balance loss will continue through the end of the century.

We have included the following sentence in the introduction of the manuscript:

P2 L18-20 “60% of this recent increase in GrIS sea level contribution is due to enhanced surface runoff [3, 5], and GrIS surface processes will also play an important role in a warming climate [2].”.

2. I am a bit concerned about the greater temperature rise in the Arctic in CMIP6 than in CMIP5. I guess these are annual values, but more interesting in terms of Greenland mass balance is the

seasonality of these temperature difference. That is needed in order to really understand the doubling of surface melting during the 21st century.

We agree with this assessment and have therefore included new figures that show the seasonality of the warming, the influence of seasonal differences in the GrIS surface energy budget and differences in sea ice extent between SSP585 and RCP8.5. Please also see our more detailed answer above.

3. In the introduction I think it would be good to also discuss why you do the downscaling and run MAR instead of use the output directly from the CMIP5/6 models to assess changes in surface melt.

We have amended the following sentence in the Introduction to now read (changes in bold):

P3 L43-48 “Using a regional climate model (MAR [2, 35, 36, 37, 38]) - **that explicitly models important polar processes such as the surface mass balance, snow properties and radiative transfer** - to downscale 6 CMIP5 RCP8.5 and 5 CMIP6 SSP585 GCM projections **to a higher spatial resolution**, we find a cumulative increase of 21st century GrIS melt by +28500 Gt (+7.9 cm sea level equivalent (SLE) until 2100) and on average a 22 days longer melt season in the CMIP6 simulations.”

4. At first reading the main body I didn't understand why you didn't use all the models for Figure 2 and instead used a subset. It's explained in the supplemental material but maybe just mention in the main body that you were limited by the number of models that supplied 6 hourly fields needed by MAR.

We agree. We have included “We selected these 5 models from the CMIP6 ensemble based on the availability of 6-hourly outputs required for the dynamical downscaling in MAR.” in the main part of the manuscript (**P6 L78-80**).

5. It is curious why the mass loss deviates already in 2020 as the temperature differences happen a few decades later. This needs some discussion and perhaps it relates to seasonal temperature differences which is something you are not showing here. The timing of when the melt season starts and ends is important here and I feel this should form part of your investigation. You do mention later that there are differences in the ablation season but you don't show the temperatures as a function of month which I think would be helpful here to see how those differ between CMIP5/6.

We now show seasonal differences in temperature in Figure 2.

In the added Figure 5, where we reconstruct the surface mass balance across all GCMs from the CMIP5 and CMIP6 emission scenarios, we also see this deviation around 2020 in the extreme high-emission scenario case (Fig 5A). However, here we only use the correlation between the 2m temperature around Greenland from the GCMs and the surface mass balance of the RCM. Therefore, because we only use the local Greenland temperature from the GCMs to create this figure, it is very likely a local temperature driven process not notable in the global or Arctic-wide figures. So we think that this is a temperature effect, but a more local one which is not visible in Fig.1 or Fig.2 because we either focus on global or Arctic temperature anomalies there. We have included the following sentence in the results section of the manuscript:

P15 L168-170 “The surface mass balance in RCP8.5 and SSP585 already starts to deviate around 2020, indicative that the onset of the greater atmospheric warming around Greenland in SSP585 lies in the early 21st century.”

6. I don't understand the statement on page 8 “The CMIP6 forced MAR simulations therefore do not have an inherent bias in SMB”. How have you shown this? You end that paragraph with a

comment on the ECS but you don't explain how that could explain what you have discussed in that paragraph. I feel this entire section needs to be expanded upon.

We agree with the reviewer that on this figure alone we can't say that MAR SSP585 doesn't have an inherent bias in SMB. However, in the methods section we also discuss and evaluate the MAR simulations against MAR forced by reanalysis for 5 essential Greenland climate variables over the present-day climate, and together with the presented figure here we see no evidence for an inherent bias over the current climate in MAR. We have tried to expand and reshuffle this paragraph and now also reference to the longer discussion in the Methods section of the manuscript.

P10 L108-122 “The surplus in SSP585 GrIS surface mass loss only becomes notable from 2020 onwards. During the period of stable SMB 1961-1990, the SSP585 simulations on average have a +26 Gt/yr greater annual SMB, a difference of around 5% when compared to the annual SMB. Conversely,

at the end of the century (2071-2100 average) the SMB is -602 Gt/yr lower in our SSP585 simulations compared to the RCP8.5 simulations, with the differences reaching up to 292% during 2090-2100. Therefore, the SSP585 simulations acquire their differences during the course of the 21st century. This is also highlighted by our comparison of MAR forced by reanalysis to which is also highlighted by our comparison of MAR forced by reanalysis to each individual MAR simulation forced by GCMs over the current climate in the Methods section (Supplementary Fig. S1-S5). Fig. S1-S5 show that over the current climate the SMB and other variables in MAR forced by reanalysis compared to GCM forcing do not differ significantly. Over the current climate (1981-2010) the SMB in MAR forced by SSP585 models does not show an inherent bias compared to the MAR RCP8.5 simulations (mean bias: 29.8 mm w.e. (SSP585) compared to 26.3 mm w.e. (RCP8.5)). Therefore, the most likely explanation for greater GrIS surface mass loss is the higher equilibrium climate sensitivity (ECS) in CMIP6 models and a more pronounced Arctic amplification (Fig. 1, 2) [29, 32, 33].”

7. On page 10 you say the melt season extends, but this 22 days, is that by the end of the century? That doesn't really seem like a large extension in the melt season. In the Arctic Ocean places already have extended melt season of more than 6 weeks between 1980 and today.

Yes, but the melt season extent is severely limited by the availability of solar radiation over the Greenland Ice Sheet, given that observational studies clearly show that absorbed shortwave radiation is the main driver of surface melt over the dark ablation zone (e.g. V.d.Broeke (2011)). Therefore, the melt is mostly *intensified* during the melt season by increasing temperature in our simulations, rather than an extensively prolonging the melt season. By comparison, most of the extra melt is due the intensification of mass loss during the already existing ablation period.

Additionally, sea ice has two major heat sources which act on very different time scales due to their difference in heat capacity. The atmosphere above and the ocean below, therefore the two systems are not directly comparable.

8. And are you also saying on page 10 that there are no differences in winter accumulation between CMIP5/6? And that winter temperatures are similar? Arctic amplification tends to be most prominent in autumn and winter, not in summer, so this is counter to what you are suggesting, in that you seem to suggest summer temperatures are the big difference between CMIP5/6, which is not really an Arctic amplification signal related to any ice-albedo feedbacks, but rather changes in atmospheric circulation, clouds and water vapor.

We are now discussing both sides – the seasonality in warming (arctic amplification) in our forcing fields and other feedbacks throughout the manuscript (sea ice, surface energy budget and cloud optical depth). We hope that this and previous comments in this answer to reviewers will address this comment.

9. In your discussion, since you are talking about Arctic air temperatures, the longer melt season at the end of the melt is more because of the loss of sea ice and increased ocean mixed layer temperatures which delays freeze-up of the ocean and therefore keeps Arctic temperatures warmer longer. I realize you talk about changes in the melt season for the whole of Greenland but be good to see regionally where this is as there may be links to sea ice cover.

See also previous answers, but we have included a specific discussion and figure about the effect of lower sea ice cover in our CMIP6 SSP585 simulations (Fig.8). These results clearly show what Reviewer #3 suggests, in that sea ice cover might be one of the causes of the additional melt we're seeing. Based on this sea ice related suggestion and our results now also highlighting a potential radiative (cloud) feedback as a potential mechanism, we have amended also the discussion of our manuscript to

P22 L263-266 “We identify a stronger Arctic amplification signal in the CMIP6 SSP585 ensemble – together with associated sea ice (Fig. 8) and radiative (cloud) feedbacks (Fig. 7) - as the main drivers behind the increase in GrIS melt and SMB reduction, which emphasizes the need for realistic representations of high-latitude climate physics in state-of-the-art global climate models.”.

References

Van den Broeke, M. R., Smeets, C. J. P. P., & Van de Wal, R. S. W. (2011). The seasonal cycle and interannual variability of surface energy balance and melt in the ablation zone of the west Greenland ice sheet. *The Cryosphere*, 5, 377-390.

Reviewers' Comments:

Reviewer #3:

Remarks to the Author:

I am generally happy with the revisions the authors have made to the paper. I think the outstanding issue remains some discussion on how realistic the larger equilibrium climate sensitivity in CMIP6 compared to CMIP5 is. While some of this may be a result of improved cloud and aerosol modeling, it is not clear if these are translating into a more realistic estimate of ECS. Some scientists suggest such high ECS in CMIP6 are inconsistent with evidence from palaeoclimate records. I know that for sure the HadUKGEM model has very high ECS and in fact loses its winter sea ice cover by the end of the century. So does IPSL. Thus, I do wonder how realistic these melt estimates really are. It may be good to provide a table with the ECS for each model used and how those numbers compare to CMIP5. A caveat is needed as I do not believe we know enough to say with 100% certainty that these high ECS are correct. I remain concerned that all projections from CMIP6 are too warm and thus, we have to be careful making these statements that so much more Greenland melt will occur.

REVIEWERS' COMMENTS

Reviewer #3 (Remarks to the Author):

I am generally happy with the revisions the authors have made to the paper. I think the outstanding issue remains some discussion on how realistic the larger equilibrium climate sensitivity in CMIP6 compared to CMIP5 is. While some of this may be a result of improved cloud and aerosol modeling, it is not clear if these are translating into a more realistic estimate of ECS. Some scientists suggest such high ECS in CMIP6 are inconsistent with evidence from palaeoclimate records. I know that for sure the HadUKGEM model has very high ECS and in fact loses its winter sea ice cover by the end of the century. So does IPSL. Thus, I do wonder how realistic these melt estimates really are. It may be good to provide a table with the ECS for each model used and how those numbers compare to CMIP5. A caveat is needed as I do not believe we know enough to say with 100% certainty that these high ECS are correct. I remain concerned that all projections from CMIP6 are too warm and thus, we have to be careful making these statements that so much more Greenland melt will occur.

We would like to thank Reviewer #3 for taking time to review our revised manuscript and for the additional input on “how realistic CMIP6 ECS estimates are compared to CMIP5”. Overall, we wanted to highlight that we don’t think that our manuscript states that we think that CMIP6 SSP585 is any shape more “likely” to manifest itself than CMIP5 RCP8.5. Given the great uncertainties whether we will follow a high- mid- or low-emission future wouldn’t allow to assess whether SSP585 or RCP8.5 is more “realistic” in the first place. However, given the comment above, we tried to add some nuance to our text and discussion in the following way:

L37-L40 (Introduction). While CMIP6 models are based on more sophisticated physics and are run at a higher resolution than its predecessor CMIP5, this study does not aim to suggest whether a CMIP6-based world with a greater sensitivity to greenhouse gas emissions comprises a more likely future scenario than CMIP5-based estimates.

L270-275 (Discussion). However, first analysis suggests, despite more sophisticated physics in CMIP6, that in individual models (CESM2) of the CMIP6 ensemble the climate sensitivity might not be in-line with paleoclimate records (i.e. too high) [47]. A similar analysis has not been done across all CMIP6 models and therefore a thorough assessment of the models' ability to realistically reproduce paleoclimatic warming periods cannot be assessed at this stage.

Additional new material:

Table S1 Showing the Equilibrium Climate Sensitivity across all our 6 CMIP5 models and the CMIP5 mean ECS.

Table S2 Showing the Equilibrium Climate Sensitivity across all our 5 CMIP6 models and the CMIP6 mean ECS.

References

[47] Zhu, J., Poulsen, C. J., and Otto-Bliesner, B. L.: High climate sensitivity in CMIP6 model not supported by paleoclimate, *Nature Climate Change*, 10, 378–379, 2020.